# CONTROLLABLE FIRST-FRAME-GUIDED VIDEO EDITING VIA MASK-AWARE LoRA FINE-TUNING

**Chenjian Gao**[1], **Lihe Ding**[1], **Xin Cai**[1], **Zhanpeng Huang**[†2], **Zibin Wang**[2], **Tianfan Xue**[†1,3,4]

[1]Multimedia Laboratory, The Chinese University of Hong Kong; [2]SenseTime Research
[3]Shanghai AI Laboratory; [4]CPII under InnoHK
`{gc025, dl023, cx023, tfxue}@ie.cuhk.edu.hk;`
`{soaroc.huang, wangzb02}@gmail.com`

## ABSTRACT

Video editing using diffusion models has achieved remarkable results in generating high-quality edits for videos. However, current methods often rely on large-scale pretraining, limiting flexibility for specific edits. First-frame-guided editing provides control over the first frame, but lacks fine-grained control over the edit's subsequent temporal evolution. To address this, we propose a mask-based LoRA (Low-Rank Adaptation) tuning method that adapts pretrained Image-to-Video models for flexible video editing. Our key innovation is using a spatiotemporal mask to strategically guide the LoRA fine-tuning process. This teaches the model two distinct skills: first, to interpret the mask as a command to either preserve content from the source video or generate new content in designated regions. Second, for these generated regions, LoRA learns to synthesize either temporally consistent motion inherited from the video or novel appearances guided by user-provided reference frames. This dual-capability LoRA grants users control over the edit's entire temporal evolution, allowing complex transformations like an object rotating or a flower blooming. Experimental results show our method achieves superior video editing performance compared to baseline methods. The code and video results are available at our project website: https://cjeen.github.io/LoRAEdit.

## 1 INTRODUCTION

Recent advances in diffusion models (Rombach et al., 2022; Lipman et al., 2023) have demonstrated unprecedented improvement in high-quality video generation (Yang et al., 2025b; Kong et al., 2024; Wang et al., 2025; HaCohen et al., 2024). Based on foundational video generation models, video editing has experienced dramatic improvement (Jiang et al., 2025; Hu et al., 2025), now finding wide application in scientific and creative fields (Yang et al., 2024). Still, these video editing models often require computationally intensive finetuning, with a large set of training data. This makes them expensive to extend to a new editing type, and less flexible for new applications. In contrast, first-frame-guided video editing (Ouyang et al., 2024; Ku et al., 2024) offers a promising path toward flexible video manipulation. In this paradigm, users can edit the first frame arbitrarily, either using image AI tools or traditional editing software. These edits are then propagated to the entire sequence, enabling flexible video manipulation without being constrained by dataset-specific training.

While first-frame-guided solutions allow flexible editing, they only provide limited control of remaining frames. For instance, given a video of a blooming flower, the user can edit the flower in the first frame, but cannot control how the flower blooms in the following frames. Similarly, when an object rotates to a novel viewpoint, the user cannot control the disoccluded region. In addition, the first frame edits may diffuse into unedited regions, resulting in undesirable background changes. The inability to control later frames limits editing flexibility and necessitates methods that not only retain the flexibility of first-frame-guided editing, but support control throughout the video.

A simple solution is per-video finetuning of a pre-trained image-to-video (I2V) model (Kong et al., 2024; Wang et al., 2025). By finetuning the model using LoRA (Hu et al., 2022) on a source video,

---

[†]Corresponding authors.

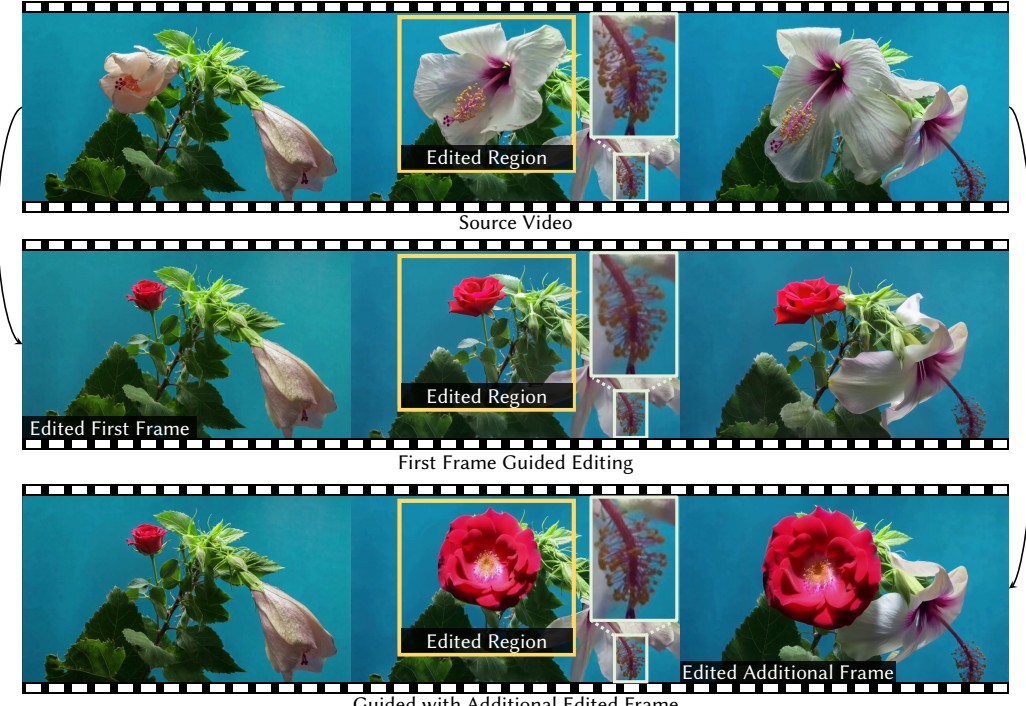

Figure 1: Given a source video (top row), we achieve high-quality video editing guided by the first frame as a reference image (middle row), while maintaining flexibility for incorporating additional reference conditions (bottom row).

the model will learn content motion. This allows the edit to propagate in a temporally consistent way. However, this naive approach lacks finer control—it cannot distinguish between regions that should change and those that should stay, nor does it ensure the edited region's appearance remains controllable as it moves and deforms over time, requiring the synthesis of unseen appearance.

In this work, we build a flexible video editing model by expanding this naive edit propagation approach with an additional mask, which controls which regions of the video remain unchanged and which are modified. Recent I2V models (Kong et al., 2024; Wang et al., 2025) are designed to generate videos from a single image, but they can also process video sequences, with a built-in masking mechanism to control which regions are preserved or modified during inference. Typically, this is only a temporal mask, which preserves the first frame while generating the subsequent frames. However, we observe the mask has greater potential for more precise spatial control over video content. To leverage this, we apply LoRA to fine-tune the model on the input video with the edited region masked. This allows the model to learn how to interpret a flexible spatiotemporal mask as a command to preserve content from the source video or generate new content in designated regions. After LoRA training, the model can effectively apply the mask, leaving unedited areas unchanged.

More importantly, our mask-guided control empowers LoRA to learn selectively, adapting to the specific demands of the edit. This flexibility is illustrated by the blooming flower example in Fig. 1. First, by configuring the mask for motion learning, LoRA learns the flower's blooming motion from the source video (second row). Then, to control the flower's appearance as it blooms, the mask is reconfigured for appearance learning, directing LoRA to capture the target appearance from an additional reference, such as the bloomed, vibrant red rose (third row). This dual capability allows our method to synthesize a controllable transformation, creating a video where the flower not only moves correctly but evolves into the desired state, a feat unattainable with naive first-frame guidance.

Our approach offers a simple and effective solution for video editing by leveraging LoRA's capabilities, without modifying the model architecture, and maintaining high flexibility through the combination of different conditions. Experimental results demonstrate that our method achieves superior performance over state-of-the-art approaches in both qualitative and quantitative evaluations.

## 2   Related Works

**Video Editing with Diffusion Models.** The success of video diffusion models has spurred extensive research into video editing. Early works adapt the image diffusion network and training paradigm to video generation and editing. Tune-A-Video Wu et al. (2023) explores the concept of one-shot tuning in video editing. Fairy Wu et al. (2024) edits keyframes utilizing a 3D spatio-temporal self-attention extended from a T2I diffusion model. VidToMe Li et al. (2024) introduces image editing approaches (e.g., ControlNet Zhang et al. (2023)) to video generation. Animatediff Guo et al. (2023) decouples the appearance and motion learning during video editing. SAVE Song et al. (2024) chooses to fine-tune the feature embeddings that directly reflect semantic information. Another line of work manipulates the hidden features to edit a video. Video-P2P Liu et al. (2024) and Vid2Vid-Zero Wang et al. (2023) employ cross-attention map injection and null-text inversion for video editing. TokenFlow Geyer et al. (2023) leverages motion-based feature injection, and FLAT-TEN Cong et al. (2023) further introduces optical flow for better injection. Other methods Chen et al. (2023); Yang et al. (2023) explore latent initialization and latent transition in video diffusion models. Dragvideo Deng et al. (2024) achieves interactive drag-style video editing by introducing point conditioning. Recently, VACE Jiang et al. (2025) has shown promising video editing ability by large-scale conditional video diffusion training. Although large video editing diffusion models achieve impressive results, they often struggle with inaccurate identity preservation and suboptimal performance on out-of-domain test cases. In contrast, our method effectively leverages powerful video priors while efficiently learning content from both the reference image and the source video.

**First-Frame Guided Video Editing.** First-frame guided editing has emerged as a mainstream video editing approach, with AnyV2V Ku et al. (2024) and I2VEdit Ouyang et al. (2024) as representative methods. These approaches decompose video editing into two stages: (i) editing the first frame using existing image methods, and (ii) propagating edits to remaining frames using motion-conditioned image-to-video diffusion models. AnyV2V reconstructs motion via DDIM sampling, injecting temporal attention and spatial features from the original video. I2VEdit enhances this by learning coarse motion through per-clip LoRA and refining appearance using attention difference masks. While this decoupled framework benefits from advances in both image editing and video generation, the lack of explicit constraints often leads to diluted edits during propagation, manifesting as foreground inconsistencies and background leakages.

## 3   Method

In this work, we introduce a controllable first-frame-guided video editing method based on recent image-to-video diffusion models (Wang et al., 2025; Kong et al., 2024). In Sec. 3.1, we first tackle the issue of maintaining coherent motion of the edit by using LoRA to transfer motion patterns from the input video. In Sec. 3.2, we explore the generalization capabilities of the mask-based conditioning mechanism in pretrained I2V models. In Sec. 3.3, we demonstrate how mask-aware LoRA enables flexible video editing by leveraging the mask to control the generated content.

### 3.1   LoRA's First Step: A Simple Solution for Video Editing

In this section, we introduce a naive approach for edit propagation, which serves as a foundation for the subsequent improvements. Given an input video $\mathbf{V}_{\text{input}} = [\mathbf{I}_1, \mathbf{I}_2, \ldots, \mathbf{I}_T]$ and an edited version of the first frame, $\tilde{\mathbf{I}}_1$, the goal is to generate an edited video $\tilde{\mathbf{V}}_{\text{edited}} = [\tilde{\mathbf{I}}_1, \tilde{\mathbf{I}}_2, \ldots, \tilde{\mathbf{I}}_T]$ where the edits introduced in $\tilde{\mathbf{I}}_1$ are propagated across all subsequent frames with coherent motion.

To achieve this basic objective, we insert LoRA (Hu et al., 2022) modules $\phi_\theta$ into the self-attention and cross-attention layers of the I2V model(Wang et al., 2025) and optimize them on the input video $\mathbf{V}_{\text{input}}$ to capture its motion pattern. During training, the model is conditioned on the original first frame $\mathbf{I}_1$ and a textual prompt composed of a fixed special token $p^*$ concatenated with the caption $c$ generated for $\mathbf{I}_1$ using Florence-2 (Xiao et al., 2024) (i.e., $[p^*] + c$). The model is supervised to reconstruct the full input video $\mathbf{V}_{\text{input}} = \{\mathbf{I}_1, \mathbf{I}_2, \ldots, \mathbf{I}_T\}$. Following the flow matching objective of the I2V model (Wang et al., 2025), we optimize the LoRA adapters by minimizing the error between

the network velocity prediction and the target flow drift:

$$\mathcal{L}_{\text{flow}} = \mathbb{E}_{t, \mathbf{x}_0, \mathbf{x}_1} \left[ \| v_\theta(\mathbf{x}_t, t; \underbrace{\mathbf{I}_1, [p^*] + c}_{\text{condition}}) - (\mathbf{x}_0 - \mathbf{x}_1) \|_2^2 \right],$$

$$\mathbf{x}_t = (1 - t)\mathbf{x}_1 + t\mathbf{x}_0 \tag{1}$$

where $\mathbf{x}_0 \sim \mathcal{N}(0, \mathbf{I})$ represents the sampled Gaussian noise, and $\mathbf{x}_1 = \mathcal{E}(\mathbf{V}_{\text{input}})$ denotes the latent representation of the source video encoded by the VAE $\mathcal{E}$. Here, $v_\theta$ is the velocity prediction network with LoRA parameters $\phi_\theta$, and $t \in [0, 1]$ is the time step. The term $(\mathbf{x}_0 - \mathbf{x}_1)$ serves as the flow target, guiding the interpolation from data $\mathbf{x}_1$ to noise $\mathbf{x}_0$.

At inference time, the original frame $\mathbf{I}_1$ is replaced with an edited version $\tilde{\mathbf{I}}_1$, and a new caption $\tilde{c}$ is generated for $\tilde{\mathbf{I}}_1$ using Florence-2. The prompt token $p^*$ is concatenated with $\tilde{c}$ to form the inference prompt $[p^*] + \tilde{c}$, which guides the generation of the edited sequence $\tilde{\mathcal{V}}$.

## 3.2 THE MASK'S HIDDEN POWER: EXPLORING I2V MODEL CAPABILITIES

The naive edit propagation introduced above only ensures motion coherence and lacks control over the content of subsequent frames. To address this, we leverage the conditioning mechanisms in recent I2V models (Wang et al., 2025; Kong et al., 2024). Specifically, to introduce the first frame as the guidance for video generation, these models incorporate two additional conditions for the denoising network: a pseudo-video $\mathbf{V}_{\text{cond}}$ and a binary spatiotemporal mask $\mathbf{M}_{\text{cond}}$. The pseudo-video $\mathbf{V}_{\text{cond}} \in \mathbb{R}^{C \times T \times H \times W}$ is constructed by concatenating the first frame $\mathbf{I} \in \mathbb{R}^{C \times 1 \times H \times W}$ with zero-placeholder frames. The binary mask $\mathbf{M}_{\text{cond}} \in \{0, 1\}^{1 \times T \times h \times w}$ is designed so that 1 indicates the preserved frame and 0 represents the frames to be generated, with the first frame set to 1 and all subsequent frames set to 0.

We extend this paradigm to video-to-video generation by replacing the pseudo-video condition $\mathbf{V}_{\text{cond}}$ with actual video frames, enabling the model to accept an entire video sequence as input. In this setting, the binary spatiotemporal mask $\mathbf{M}_{\text{cond}}$, originally designed to preserve only the first frame, can now be repurposed as a more flexible mechanism that selectively controls which regions are retained and which are regenerated across space and time.

To assess the generalization capabilities of the masking mechanism, we evaluate several binary mask configurations, as shown in Fig. 2. In each case, the mask $\mathbf{M}_{\text{cond}}$ is applied to the input video to construct $\mathbf{V}_{\text{cond}}$, where regions marked as zero are regenerated and the rest are preserved. The default I2V configuration preserves only the first frame and leads the model to synthesize motion across the entire sequence (first row). Exploring the extremes, an all-zeros mask that preserves none of the original content forces the model to generate the appearance for the entire video (second row). Conversely, an all-ones mask aimed at preserving all content

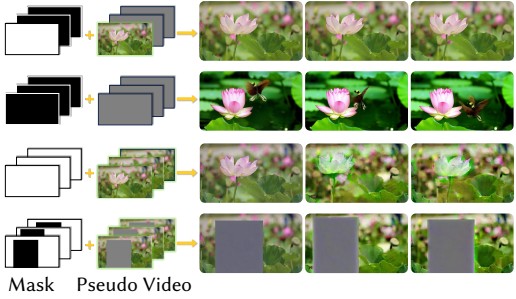

Mask    Pseudo Video

Figure 2: Exploring different mask configurations as an input condition to image-to-video model.

effectively maintains the video's overall structure but introduces artifacts in areas with discontinuous motion (third row). Finally, using a spatially varying mask to preserve the background while generating the foreground reveals a key challenge, as the model struggles to synthesize coherent foreground content (fourth row).

**Analysis and Motivation.** The preceding cases show the raw I2V model can handle simple, full-frame instructions but fails at nuanced, selective editing. Our key insight is to repurpose this mechanism, enabling the spatiotemporal mask to serve a dual purpose during LoRA tuning. First, we can use LoRA to reinforce the model's response to the mask, improving its ability to execute the preservation and generation commands defined by the mask. More critically, we can use the mask to direct what LoRA learns. By masking different content during training, we can guide LoRA to focus on either the video's underlying motion or a reference's target appearance. This interplay between LoRA and the mask is the cornerstone of our method, detailed in the following section.

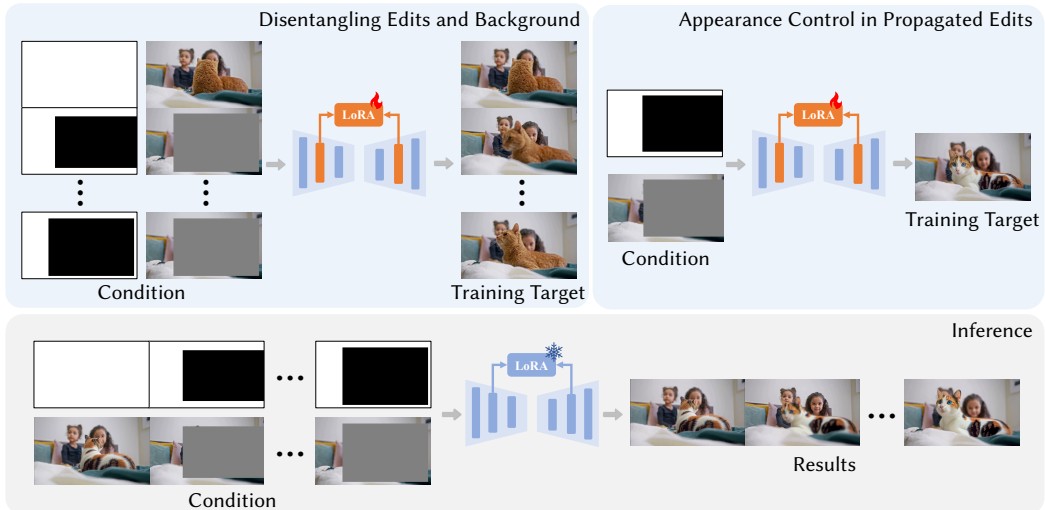

Figure 3: Our mask-guided LoRA pipeline. Training (Top): LoRA is fine-tuned to learn motion from the masked source video (left) and appearance from a reference frame (right). Inference (Bottom): The trained LoRA applies the learned motion and appearance to an edited first-frame, producing a temporally consistent video.

## 3.3 UNLOCKING EDITING FLEXIBILITY: MASK-GUIDED LORA

Building on this exploration, we modify the spatiotemporal mask to enable more flexible video edits. Combined with LoRA fine-tuning, the mask serves two complementary roles. First, it improves the I2V model's alignment with mask constraints, allowing users to limited editing to a specific region, achieving more flexible control. Second, it also acts as a signal guiding LoRA to learn specific patterns from the training input, such as motion pattern from video sequences or appearance of an object from images. Specifically, we adapt the flow matching loss in Eq. 1 to incorporate the conditioning video and mask:

$$\mathcal{L} = \mathbb{E}_{t,\mathbf{x}_0,\mathbf{x}_1}\big[\|v_\theta(\mathbf{x}_t, t; \underbrace{\mathbf{V}_{\text{cond}}, \mathbf{M}_{\text{cond}}, [p^*] + c}_{\text{condition}}) - (\mathbf{x}_0 - \mathbf{x}_1)\|_2^2\big],$$

$$\mathbf{x}_t = (1 - t)\mathbf{x}_1 + t\mathbf{x}_0 \tag{2}$$

where $\mathbf{x}_1 = \mathcal{E}(\mathbf{V}_{\text{target}})$ represents the latent of the target video, and $\mathbf{x}_0$ denotes the sampled noise. As shown in Fig. 3, by configuring $\mathbf{V}_{\text{cond}}$, $\mathbf{M}_{\text{cond}}$, and $\mathbf{V}_{\text{target}}$ in different ways, we enable flexible video editing through LoRA, detailed in the following sections.

### 3.3.1 DISENTANGLING EDITS AND BACKGROUND

Many first-frame edits alter only a part of the frame(Ku et al., 2024; Ouyang et al., 2024), creating a conflict between two demands: the edited region must evolve, while the background must remain static. When a single generative pathway handles both, they may collide. Preserving the background can stall the edit, while propagating the edit can cause unintended background changes.

To achieve separate on control on edited regions and non-edit background, we carefully adjust the spatiotemporal mask $\mathbf{M}_{\text{cond}}$ and the conditioning video $\mathbf{V}_{\text{cond}}$ during LoRA fine-tuning. The mask $\mathbf{M}_{\text{cond}}$ is set to ones for the first frame to preserve it as the reference, and for subsequent frames, $\mathbf{M}_{\text{cond}}$ is adjusted to mark unedited regions with ones (to be preserved) and edited regions with zeros (to be generated). The pseudo-video $\mathbf{V}_{\text{cond}}$ is created by applying the mask to the input video, setting the regions marked as zero in $\mathbf{M}_{\text{cond}}$ to be empty, while leaving the rest unchanged. The objective $\mathbf{V}_{\text{target}}$ is set to the input video during LoRA fine-tuning. This configuration allows the model to focus on generating the edited content while locking the unedited regions. At inference time, when editing the first frame (replacing $\mathbf{I}_1$ with $\tilde{\mathbf{I}}_1$), we use the same $\mathbf{M}_{\text{cond}}$ as during LoRA training, while $\mathbf{V}_{\text{cond}}$ has its first frame replaced by the edited version $\tilde{\mathbf{I}}_1$.

One interesting observation is that while a pre-trained I2V model struggles with selective editing, LoRA training on a single video alone learns effective mask-guided inpainting priors. We speculate this is due to the diffusion transformer processing inputs as discretized tokens, with a spatially varying mask sharing a similar token-level representation, making the adaptation straightfoward.

### 3.3.2 Appearance Control in Propagated Edits

An edit in the first frame rarely stays static: the modified region may rotate, deform, or follow its own motion trajectory (e.g., petals unfolding as a flower blooms). To make the subsequent frames look natural, the model has to infer how the edited region should appear under these evolving viewpoints and states. When the only constraint is the first frame itself, this inference is under-specified, and the edit drifts away from the user's intent. To address this, we allow users to edit any subsequent frame, providing direct guidance for how the appearance should look at specific points in time.

During LoRA fine-tuning, we use an edited frame as the target $\mathbf{V}_{target}$. The conditioning input $\mathbf{V}_{cond}$ is constructed using the pre-edited frame by masking out the edited regions. The mask $\mathbf{M}_{cond}$ marks the preserved background areas with ones and the edited regions with zeros. If multiple edited frames are used, we decouple appearance from motion by training on each frame as an isolated, static image, preventing the model from inferring false temporal dynamics between them. This configuration allows the model to learn how edited content should appear in context, guided by both the surrounding background and the user-provided modification.

Unlike methods that directly feed edited frames as inputs during inference (Yang et al., 2025a; Jamriska, 2018), we do not require the edited frame to remain the same during inference. Instead, the edited frame is used only during training to guide how edits should appear. At inference time, the model generates content based on learned patterns and context, allowing it to adapt edits smoothly across frames, even when the edits do not adhere to strict temporal alignment.

## 4 Experiments

### 4.1 Implementation Details

We conduct our experiments using videos consisting of 49 frames, with a resolution of either $832 \times 480$ or $480 \times 832$. All main results are obtained using the Wan2.1-I2V 480P model. Additional results based on HunyuanVideo-I2V are included in App. E. Our framework is built upon the publicly available diffusion-pipe codebase[1]. Details regarding our automated mask acquisition workflow are provided in App. A. For each video editing sample, we begin by training on the input video for 100 steps as described in Sec. 3.3.1. If additional edits are applied to later frames, we continue training for another 100 steps on data that includes additional modifications as described in Sec. 3.3.2. This helps the model incorporate user-specified appearance changes. We use a learning rate of $1 \times 10^{-4}$ for all experiments. Training on 49-frame videos typically requires 20 GB of GPU memory. In App. C, we describe a strategy that reduces GPU requirements.

### 4.2 Comparison with State-of-the-Arts

**Comparison with Reference-Guided Video Editing.** We compare our method with two recent reference-guided video editing approaches: Kling1.6 (KlingAI, 2025) and VACE (Jiang et al., 2025). To evaluate the performance on reference-guided video editing, we collect 20 high-quality video clips from Pexels and YouTube. Each video is paired with a reference image representing the desired edit. We use ACE++ (Mao et al., 2025) to apply the edit to the first frame for our method. Figure 4 shows visual comparison results. Compared to Kling1.6 and VACE, our method better respects the intended appearance in the edited region while preserving background content and temporal consistency.

**Comparison with First-Frame-Guided Video Editing.** We further compare our method with recent first-frame-guided video editing approaches, including I2VEdit (Ouyang et al., 2024), Go-with-the-Flow (Burgert et al., 2025), and AnyV2V (Ku et al., 2024). All baselines take the edited first frame as input and attempt to propagate the edits through the entire sequence. To ensure a fair

---

[1]https://github.com/tdrussell/diffusion-pipe

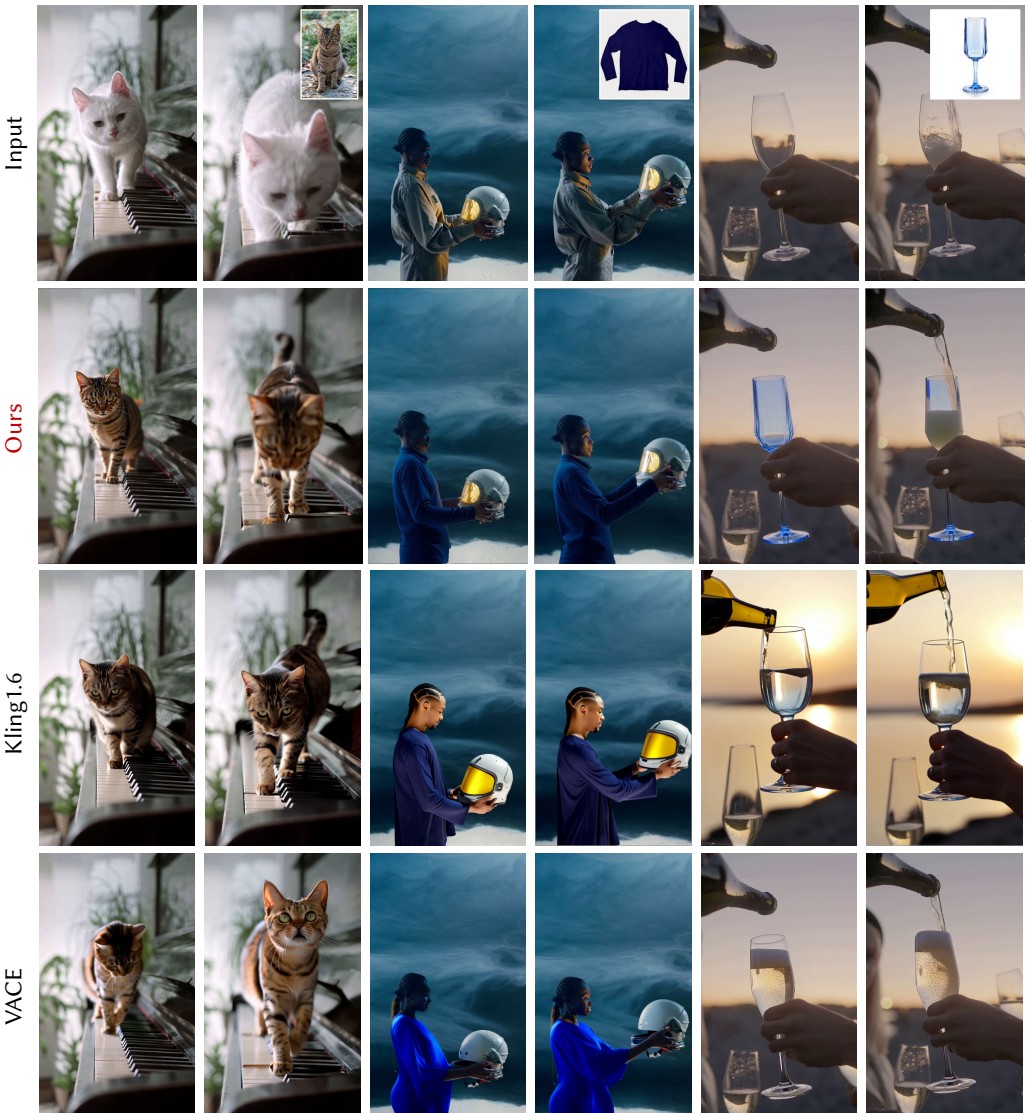

Figure 4: Comparisons with state-of-the-art reference-guided video editing methods.

and consistent evaluation, we adopt the test set from I2VEdit, which contains videos from diverse sources along with paired first-frame edits. Figure 5 shows qualitative results. In the portrait example (left), our method accurately adds the necklace while preserving the facial structure, while baseline methods often distort the face or produce artifacts. In the street scene (right), our approach transfers the clothing style cleanly across frames without affecting the background, whereas baseline methods distort the clothing or introduce changes in unedited areas.

**Quantitative Results.** For quantitative evaluation on first-frame-guided video editing, we use three metrics: 1) DeQA Score You et al. (2025), a state-of-the-art method for assessing image quality; 2) CLIP Score, which measures the semantic alignment between generated frames and edited first frame by comparing their CLIP Radford et al. (2021) embedding similarity; and 3) Input Similarity, which computes the CLIP embedding similarity between the generated frames and the input frames on a per-frame basis. As shown in Tab. 1, our method outperforms others across all metrics. For quantitative evaluation on reference-guided video editing, we conducted a user study with 35 participants. Each participant was randomly shown 10 groups of results generated by different methods.

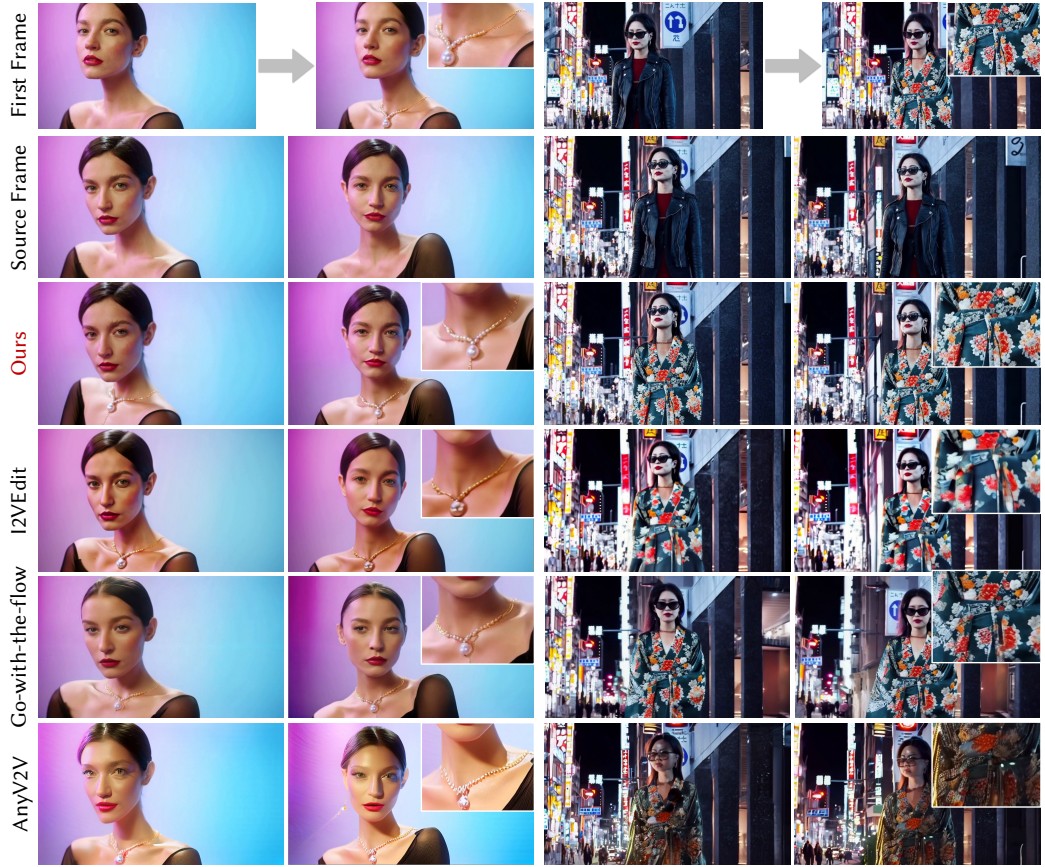

Figure 5: Comparisons with state-of-the-art first-frame-guided video editing methods.

Table 1: Quantitative comparison with first-frame-guided video editing.

|  | CLIP Score ↑ | DEQA Score ↑ | Input Similarity ↑ |
|---|---|---|---|
| AnyV2V | 0.8995 | 3.7348 | 0.7569 |
| Go-with-the-Flow | 0.9047 | 3.5622 | 0.7504 |
| I2VEdit | 0.9128 | 3.4480 | 0.7536 |
| **Ours** | **0.9172** | **3.8013** | **0.7608** |

Table 2: Average user ranking results for comparison with reference-guided video editing.

|  | Motion Consistency ↓ | Background Preservation ↓ |
|---|---|---|
| Kling1.6 | 1.869 | 1.806 |
| VACE (14B) | 2.511 | 2.460 |
| **Ours** | **1.620** | **1.734** |

For each group, the participants were asked to rank the results based on motion consistency and background preservation. Tab. 2 demonstrates the superiority of our method in both aspects.

## 4.3 ABLATION STUDIES

**Disentangling Edits and Background.** To validate the effectiveness of mask conditioning in separating edited regions from preserved content, we conduct an ablation study comparing our method using a foreground-background mask against a baseline version without it. Figure 6 shows the results. On the left, the goal is to apply a hair color change. Without mask conditioning, the edit is applied globally, altering the lighting across the frame. In contrast, with mask conditioning, the model localizes the change to the hair region while leaving the background untouched. Similarly, in the right example, clothing edits are confined to the target area only with mask conditioning.
**Appearance Control in Propagated Edits.** We conduct an ablation to evaluate the impact of using edited frames beyond the first frame for controlling appearance in edits propagation. Figure 7

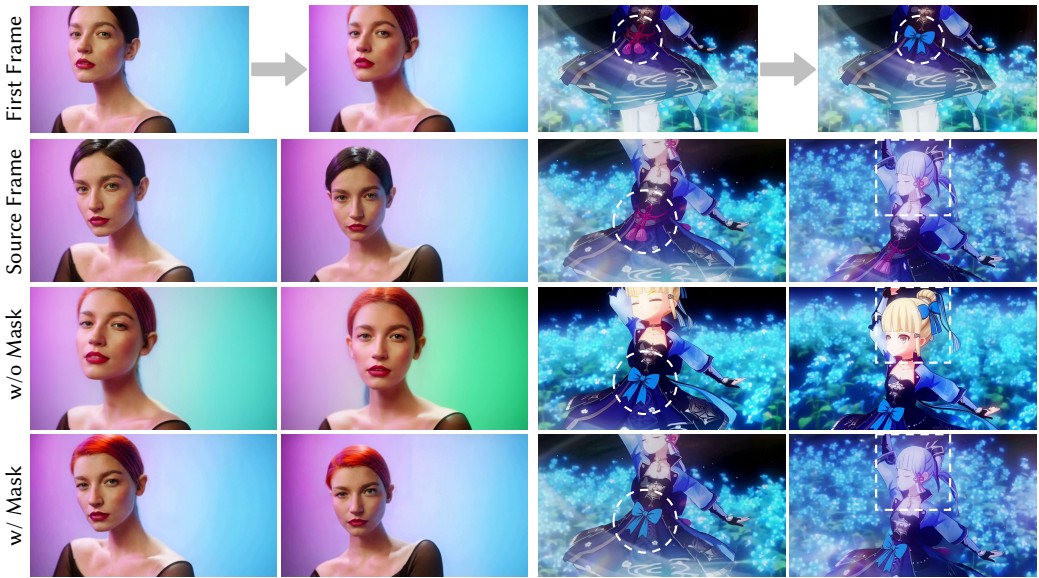

Figure 6: Ablation of disentangling edits and background.

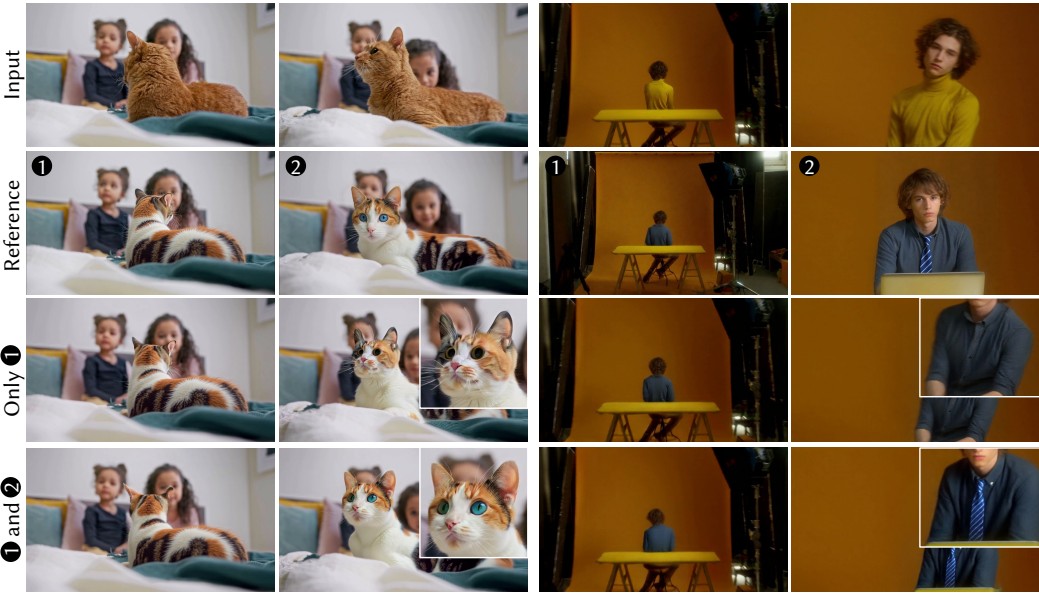

Figure 7: Ablation of incorporating additional reference.

compares two settings: using only the first frame as input versus adding an edited frame at a later timestep. While using only the first frame can still generate reasonable results, incorporating an additional edited frame offers stronger control over the appearance, leading to more consistent and accurate propagation of the intended edit.

**Mask Robustness.** To evaluate the sensitivity of our method to mask quality, we conducted an ablation study comparing results under three mask configurations: a tight mask using high-precision segmentation directly from SAM2 (Ravi et al., 2024); a noisy mask simulated by downsampling the segmentation to a $7 \times 7$ grid to introduce significant boundary errors; and the bounding box mask we used, which employs a loose rectangular region derived from the segmentation. The comparisons in Figure 8 reveal a key insight: *pixel-perfect precision is often unnecessary and can be restrictive*

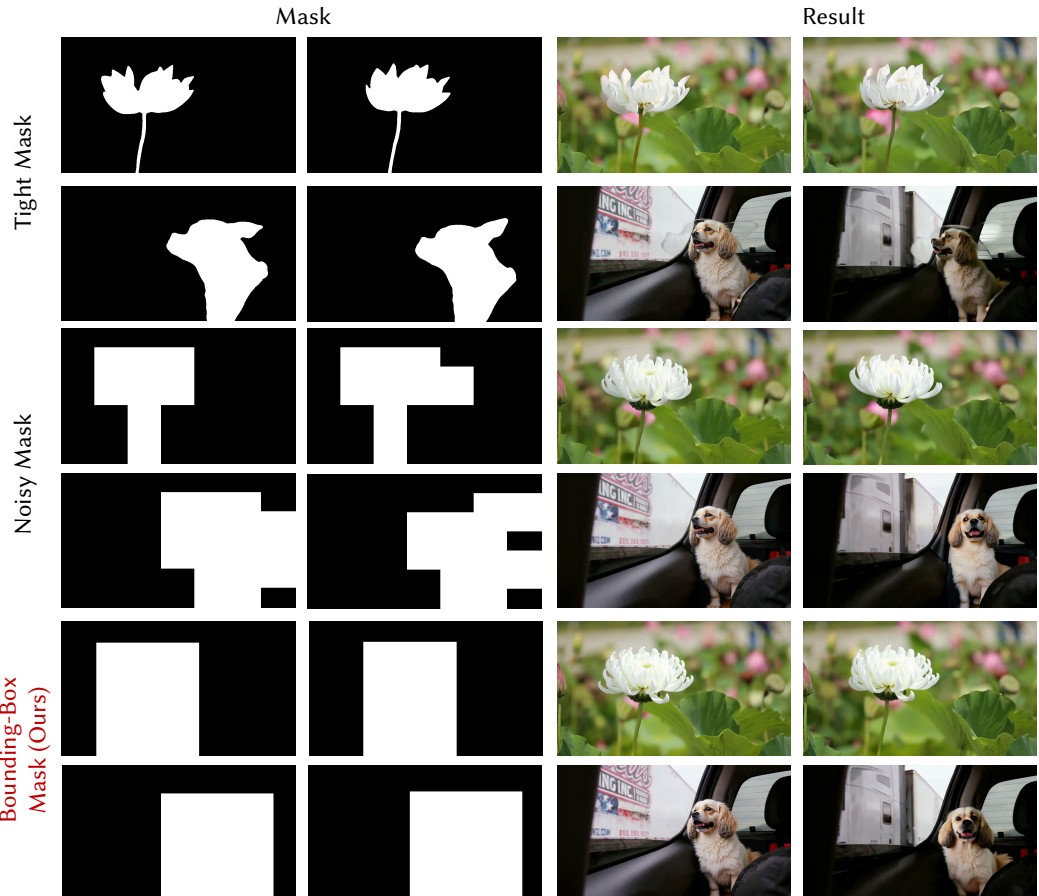

Figure 8: Ablation on mask quality. Comparison between tight mask, noisy mask, and our bounding-box strategy. As observed, pixel-perfect precision (tight mask) restricts the generation. Using a "loose" mask provides a necessary spatial buffer, allowing the model to synthesize natural transitions that seamlessly integrate with the background.

*for generative editing.* Since the goal is to alter an object's appearance, the generated entity requires a spatial buffer to undergo necessary contour variations. A strictly tight mask would excessively constrain the generation, forcing the new object to adhere rigidly to the original silhouette and potentially clipping natural details. In contrast, we observe that loose masks including both the highly perturbed "Noisy Mask" and our "Bounding Box" yield better visual results than tight segmentation. By conditioning on a looser region, we allow the model to utilize its strong priors to heal the boundary between the edit and the frozen background. This robustness confirms that our framework relies on the mask for semantic localization rather than strict pixel clipping, validating our design choice of using an automated, approximate masking workflow (See App. A).

## 5 CONCLUSION

In this work, we present a controllable first-frame-guided video editing framework leveraging mask-aware LoRA fine-tuning to achieve flexible, high-quality, and region-specific video edits without modifying the underlying model architecture. Our method enables fine-grained control over both foreground and background, supports propagation of complex edits across frames, and allows for additional appearance guidance through reference images. Experiments demonstrate that our approach outperforms existing state-of-the-art methods in both qualitative and quantitative evaluations, while maintaining temporal consistency and background preservation.

## ACKNOWLEDGEMENTS

This work is supported in part by the Centre for Perceptual and Interactive Intelligence (CPII) Ltd., a CUHK-led InnoCentre under the InnoHK initiative of the Innovation and Technology Commission of the Hong Kong Special Administrative Region Government. This work is also support by RGC Early Career Scheme (ECS) No. 24209224, CUHK-CUHK(SZ)-GDSTC Joint Collaboration Fund No. 2025A0505000053. We would like to thank Yumeng Shi for her valuable suggestions on the figures and demo. We also thank the reviewers for their insightful comments.

## ETHICS STATEMENT

We have read and adhered to the ICLR Code of Ethics. Our work, centered on a controllable video editing framework, aims to advance creative, commercial, and scientific applications. We acknowledge, however, that generative video technologies can be misused for creating misleading or harmful content, such as deepfakes. Our research is intended to provide artists and creators with more flexible and precise tools, not to facilitate malicious use. We advocate for the responsible development and deployment of generative models, accompanied by robust detection mechanisms and clear content provenance standards to mitigate such risks. Furthermore, our method relies on pre-trained foundation models (Wan2.1-I2V, HunyuanVideo-I2V), which may inherit biases from their training data. While our approach does not introduce new data, we recognize that these biases could be reflected in the generated outputs. Addressing and mitigating these inherited biases remains a critical area for future research. The datasets used for evaluation are composed of publicly available videos from sources like Pexels, YouTube, and established academic benchmarks, respecting the data usage policies of these platforms.

## REPRODUCIBILITY STATEMENT

We are committed to ensuring the reproducibility of our research. All implementation details required to replicate our results are provided in the main paper and appendix. Our method is built upon the publicly available diffusion-pipe codebase, as cited in Section 4.1. We have released our specific source code, including scripts for mask-guided LoRA training and inference. The core of our experiments relies on publicly available pre-trained models, specifically Wan2.1-I2V and HunyuanVideo-I2V, which are clearly identified in Section 4.1. This section also details key hyperparameters, such as learning rates (1e-4), training steps (100+100), and hardware requirements (20GB GPU memory). Additional results and implementation strategies, including a low-memory approach, are described in the Appendix.

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

APPENDIX

## A    IMPLEMENTATION DETAILS: MASK ACQUISITION GUI

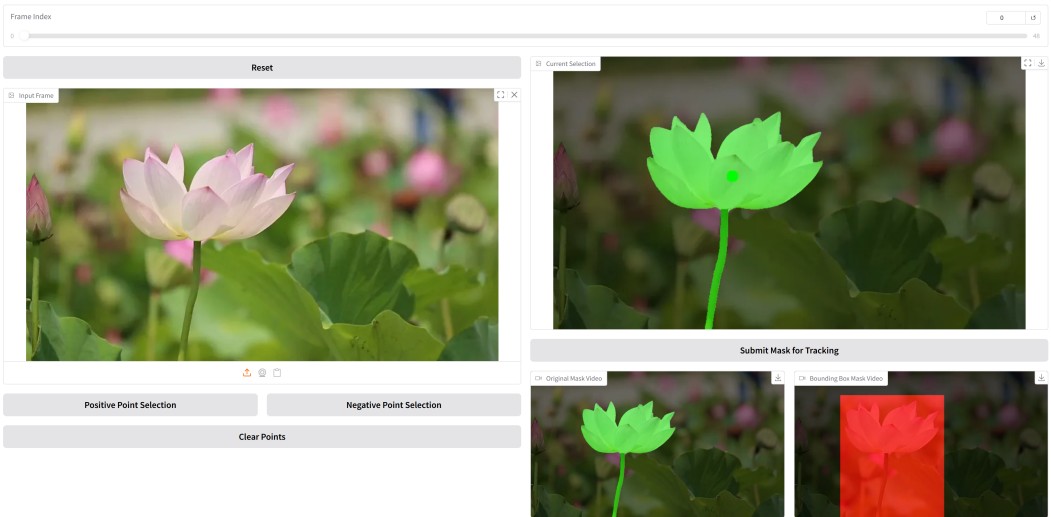

Figure 9: Screenshot of the included GUI for mask acquisition. Users initialize the process by clicking positive (green) and negative (red) points on the first frame. The segmentation is propagated via SAM2 and automatically converted into bounding box mask for training.

To facilitate easy usage, we provide a Graphical User Interface (GUI) in our codebase (see Fig. 9). Users initialize the process by providing sparse clicks on the first frame to define the target object. The mask is then propagated automatically using SAM2.

Crucially, we utilize the bounding box derived from the segmentation for training, rather than the tight mask itself. As demonstrated in Section 4.3, this loose bounding box strategy is deliberate, providing a spatial buffer for natural transitions and structural changes. This workflow confirms that our method is highly automated and does not require manual frame-by-frame annotation.

## B    ADDITIONAL EXPERIMENTAL ANALYSIS

### B.1    INPUT-LEVEL VS. FEATURE-LEVEL MASKING

To verify the gain of our mask-aware fine-tuning strategy compared to feature-level strategies which utilize masks to constrain the background at the latent feature level, we conducted a comparison.

We established a feature-level baseline where the model is fine-tuned using the default Image-to-Video mask configuration (conditioning only on the first frame, prompting the model to learn global frame reconstruction). In this setup, background preservation is enforced as an inference-time intervention. Specifically, at each denoising timestep $t$, we restrict the generation by mechanically blending the model's predicted noisy latent $z_{t-1}^{pred}$ with the noisy source latent $z_{t-1}^{src}$ using the binary segmentation mask $\mathbf{M}$ (where 1 indicates the constrained background regions):

$$z_{t-1} = z_{t-1}^{pred} \odot (1 - \mathbf{M}) + z_{t-1}^{src} \odot \mathbf{M} \tag{3}$$

This strategy forces the background pixels to remain unchanged but ignores the semantic consistency between the edited region and the strictly constrained background.

The visual comparison in Figure 10 highlights the limitation of the feature-level constraint: foreground objects often appear detached. Quantitative results are reported in Table 3. Our input-conditional approach yields consistent gains across all key metrics, confirming that providing the spatiotemporal mask as a training input enables the model to synthesize more coherent and naturally integrated edits.

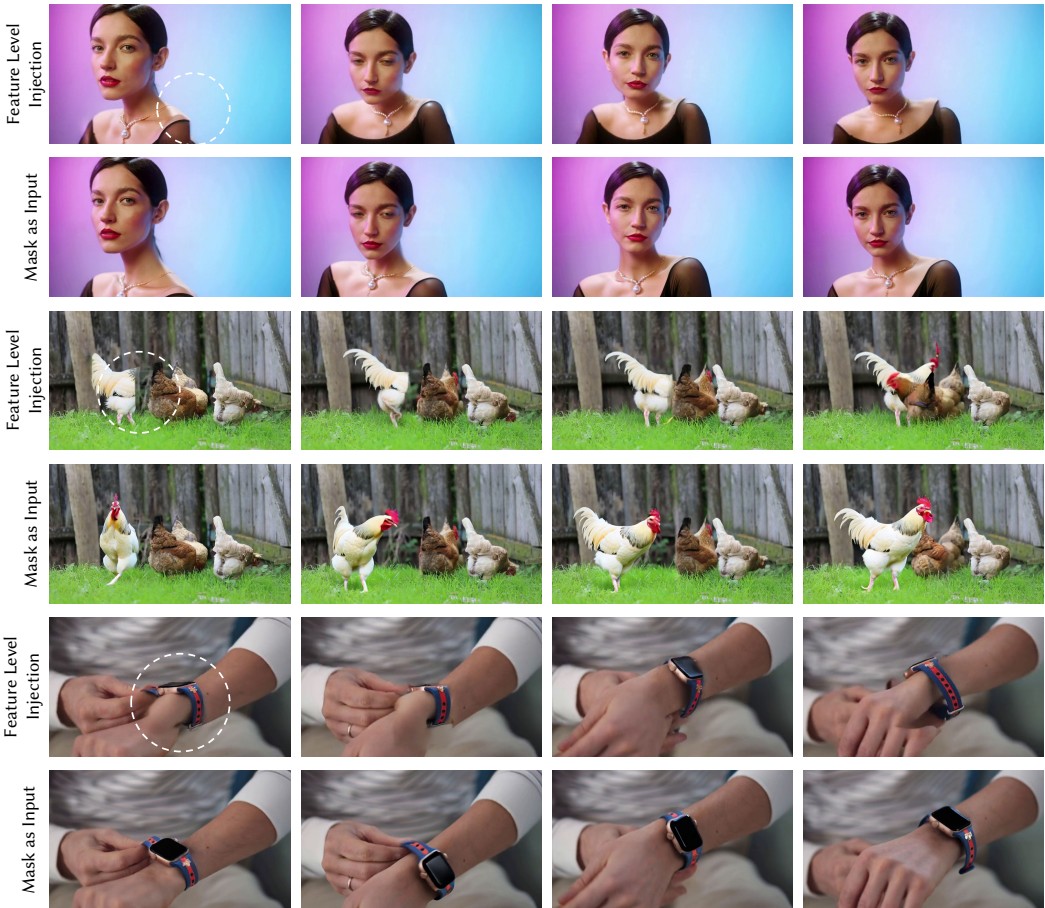

Figure 10: Visual comparison between Feature-Level masking (baseline) and our Input-Level masking strategy. As highlighted by the white circles, the Feature-Level approach often results in discontinuities and "cut-and-paste" artifacts. In contrast, our Input-Level method leverages the video prior, achieving better results.

Table 3: Quantitative comparison of masking strategies. "Feature-Level" refers to the inference-time latent blending baseline described in Eq. 3.

| Method | CLIP Score ↑ | DEQA Score ↑ | Input Similarity ↑ |
|---|---|---|---|
| Baseline (Feature-Level) | 0.8936 | 3.5878 | 0.7402 |
| **Ours (Input-Level)** | **0.9172** | **3.8013** | **0.7608** |

### B.2 EFFECTIVENESS OF MASK SCHEDULING FOR APPEARANCE LEARNING

We further validate the role of the mask during the appearance learning stage. As demonstrated in Figure 11, using the standard I2V masking configuration (without our specific region-aware scheduling) fails to effectively update the object's appearance.

This failure stems from the underlying mechanism of Image-to-Video models: the first frame is typically provided as a strong input condition. When fine-tuning on a reference image without masking, the model encounters a task where the input condition (the reference) is identical to the target output. Consequently, the optimization finds a trivial solution: it learns to simply copy the pixel information from the condition to the output. Because the appearance information is "leaked" through the input condition, the LoRA parameters do not internalize the visual attributes of the new object. Our proposed mask scheduling resolves this by masking the target region, thereby blocking

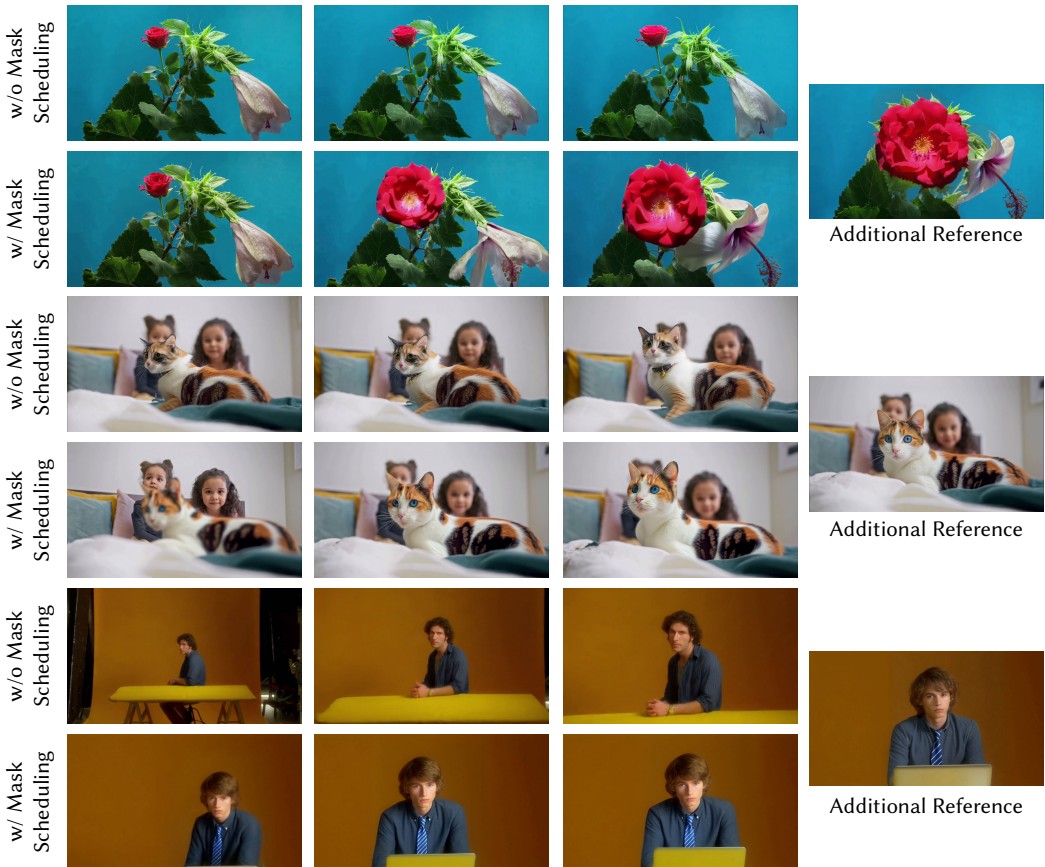

Figure 11: Ablation on mask conditioning during Stage 2 (Appearance Learning). We compare our strategy with a baseline using the default I2V mask configuration. Without our specific mask scheduling, the model exploits the first-frame condition as a shortcut, simply copying the reference image to the output without encoding the appearance into the learnable LoRA weights.

this trivial shortcut and acting as a strict command that forces the LoRA to explicitly learn and generate the target appearance distribution.

### B.3 SCALABILITY ANALYSIS

To verify the scalability and temporal stability of our method as requested, we evaluated the editing performance across video sequences of varying lengths: 5, 13, 21, and 49 frames. We utilize the CLIP Score as the metric, which measures the semantic alignment between generated frames and edited first frame by comparing their CLIP Radford et al. (2021) embedding similarity. As illustrated in Figure 12, our method demonstrates good stability.

## C EFFICIENCY ANALYSIS & LOW-COST STRATEGY DETAILS

### C.1 METHODOLOGICAL DESIGN: TEMPORAL WINDOWING

The high VRAM requirement (∼20GB) of our standard training setting primarily stems from the necessity to process the full 49-frame sequence simultaneously during training. To address this, our "Low-Cost Strategy" introduces a training-time modification that circumvents this bottleneck without altering the inference pipeline. Specifically, we split the training video into shorter, overlapping sliding windows and update the LoRA weights based on local 9-frame segments rather than the entire sequence at once.

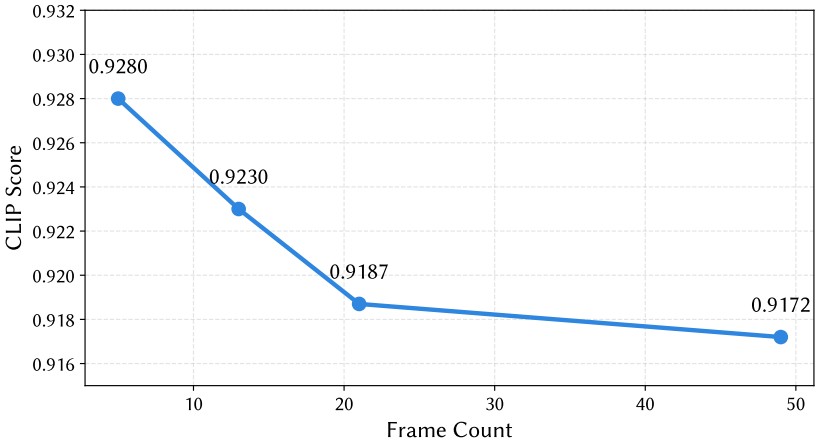

Figure 12: Frame-count vs. CLIP Score. The curve illustrates the consistency with the edited first frame across varying sequence lengths.

This design is theoretically grounded in the insight that motion LoRA primarily learns local dynamics. Since complex motions can be decomposed into continuous short-term patterns, training on 9-frame clips is sufficient to capture the necessary kinematics. Crucially, this fragmentation does not compromise global consistency because of our mask-aware conditioning. Since the unmasked background is fed as a visible input context, the model effectively learns "how the foreground moves *relative* to the fixed background" within each local window. This strong contextual anchoring ensures that the learned motion remains coherent and aligned with the environment when the full video is reassembled. As shown in Figure 13, the Low-Cost strategy maintains visual fidelity, motion smoothness, and temporal consistency comparable to standard full-frame training, validating the effectiveness of our decomposition approach.

## C.2 Engineering Optimization: Swap Blocks

To further democratize access, we leveraged the "swap blocks" technique supported by the `diffusion-pipe` codebase[2]. This mechanism dynamically offloads frozen base model parameters to the CPU, keeping only trainable LoRA weights and active transformer blocks on the GPU. Table 4 quantifies the memory savings achieved by varying the `blocks_to_swap` parameter. By maximizing swapping (config 38), we reduce the peak VRAM to ∼8 GB, making the training pipeline accessible on consumer-grade GPUs.

Table 4: Effect of "Swap Blocks" optimization on Peak VRAM (using 9-frame segments).

| `blocks_to_swap` | 0 (None) | 16 | 32 (Default) | **38 (Max)** |
|---|---|---|---|---|
| Peak VRAM (MB) | 25,097 | 16,305 | 9,987 | **7,617** |

## C.3 Run-Time VRAM Comparison with Baselines

We benchmarked the run-time VRAM usage of our method against state-of-the-art inference-only baselines on a single NVIDIA RTX 4090 ($832 \times 480$, 49 frames).

As shown in Table 5, our optimized Low-Cost strategy achieves a memory footprint comparable to inference-only baselines, demonstrating broad hardware accessibility.

---

[2]https://github.com/tdrussell/diffusion-pipe

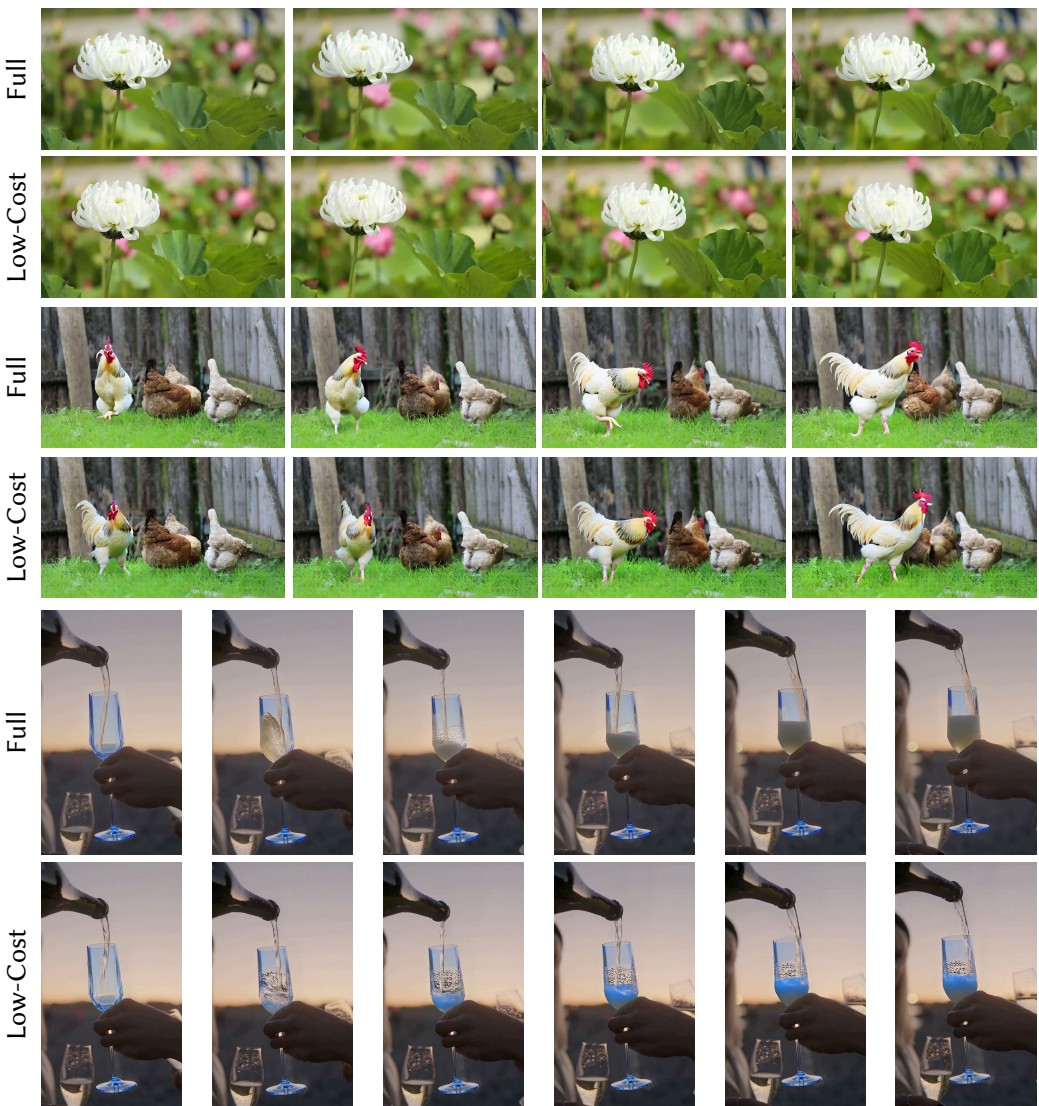

Figure 13: **Visual comparison between Standard (Full) and Low-Cost strategies.** Top rows: Results from standard training on the full 49-frame sequence. Bottom rows: Results from our optimized training using 9-frame sliding windows. The results indicate that our memory-efficient strategy preserves motion fidelity and structural details comparable to the full-frame baseline.

Table 5: Peak Run-Time VRAM comparison against inference-based methods. *For Go-with-the-flow, the first value is Sampling VRAM, second is Peak Decoding VRAM.

| Method | AnyV2V | Go-with-the-flow* | Ours (Standard) | **Ours (Low-Cost)** |
|--------|--------|-------------------|-----------------|---------------------|
| Peak VRAM | 13,680 MB | 4,398 / 21,512 MB | 21,522 MB | **7,617 MB** |

# D  DIVERSE EDITING RESULTS

We present visual results of our method on diverse editing tasks in Fig. 14 and Fig. 15, including object replacement, addition, and removal. We also include results for fast-motion and multi-person scenario in Fig. 16 and results for long-form video in Fig. 17.

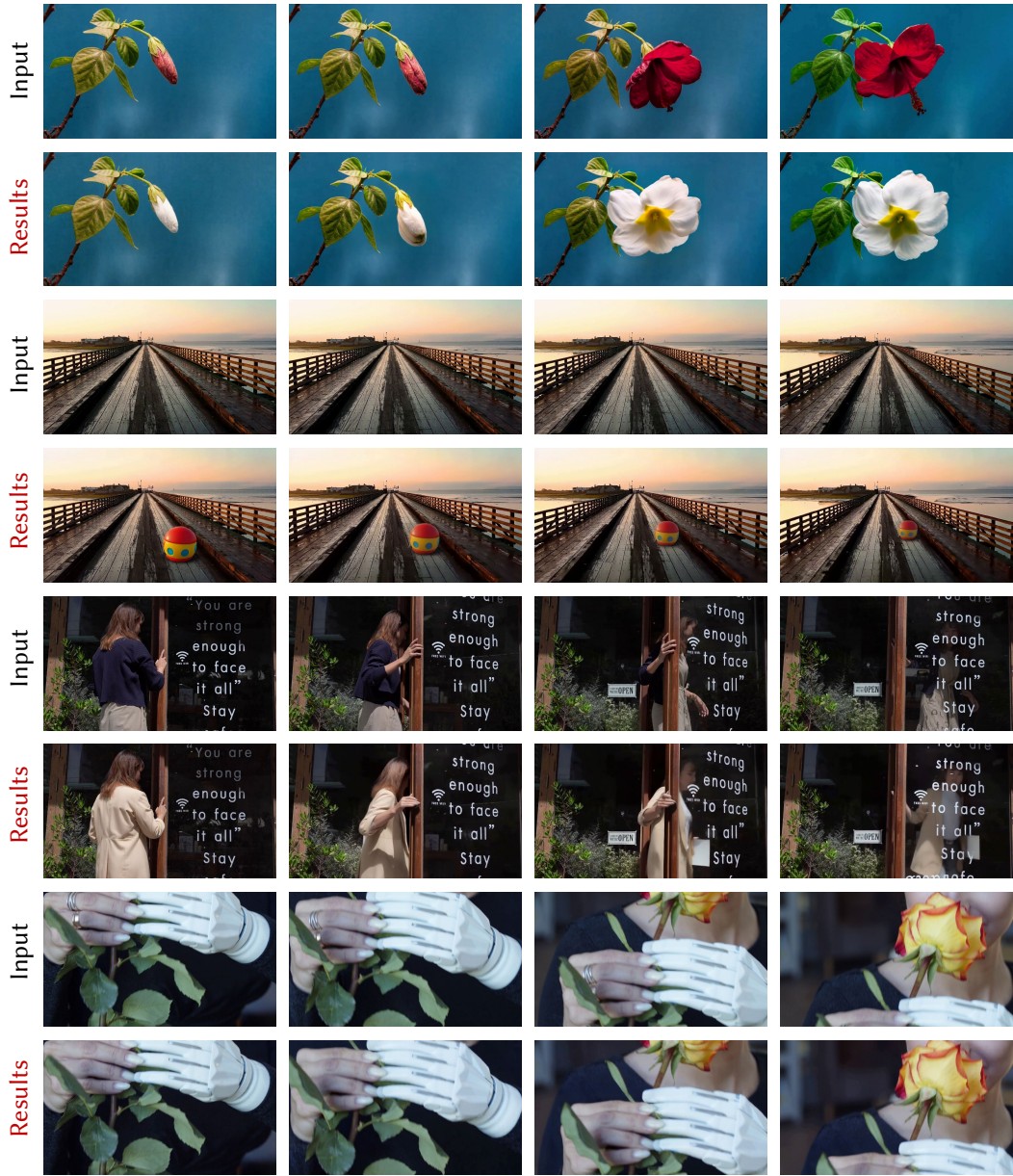

Figure 14: Diverse editing results (I).

# E  RESULTS WITH HUNYUANVIDEO-I2V MODEL

In addition to the main results based on Wan2.1-I2V, we also conducted experiments using HunyuanVideo-I2V (See Fig. 18). It demonstrates the generalization ability of our framework across different I2V architectures.

# F  FAILURE CASES

Figure 19 illustrates representative limitations encountered by our framework. We observe specific difficulties in text generation and preservation (as seen in the top rows). Even with mask constraints, the model often degrades high-frequency semantic symbols into blurred texture, likely due to the

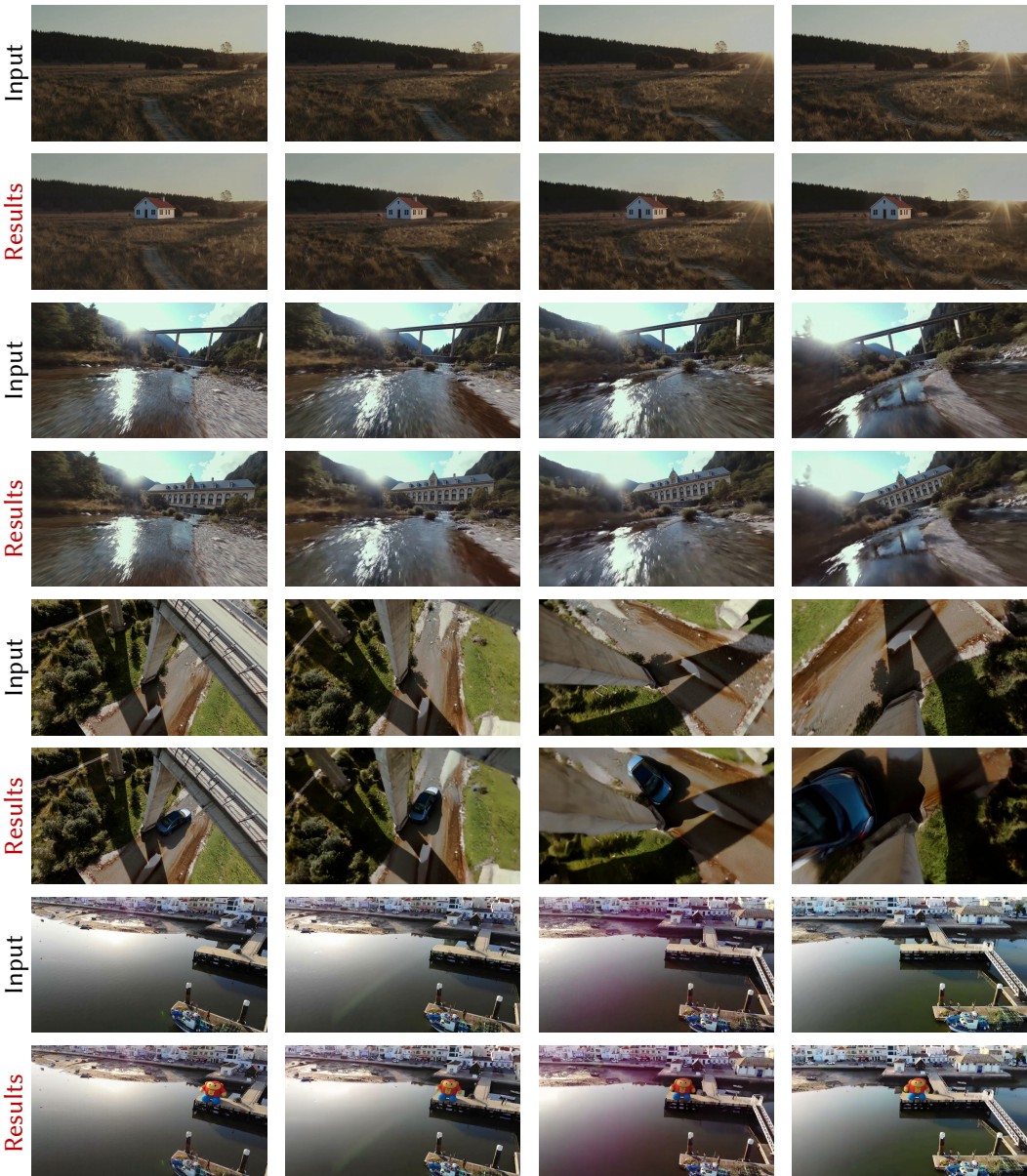

Figure 15: Diverse editing results (II).

compression loss of the underlying VAE. Additionally, limitations arise in complex motion preservation, as demonstrated in the bottom rows where the character's hair color is modified. In scenarios featuring rapid and highly stylized abstract motion, accurately decoupling the appearance update from the original dynamics becomes challenging, potentially resulting in unnatural deformations and disruptions to the structural integrity of the motion sequence.

## G  DETAILS OF USER STUDY

To evaluate the performance of our method, we conducted a user study with 50 participants. Each participant was randomly shown 10 groups of results generated by different methods. For each group, the participants were asked to select the best result based on two criteria: reference similarity and visual quality. The user study interface is shown in Figure 20.

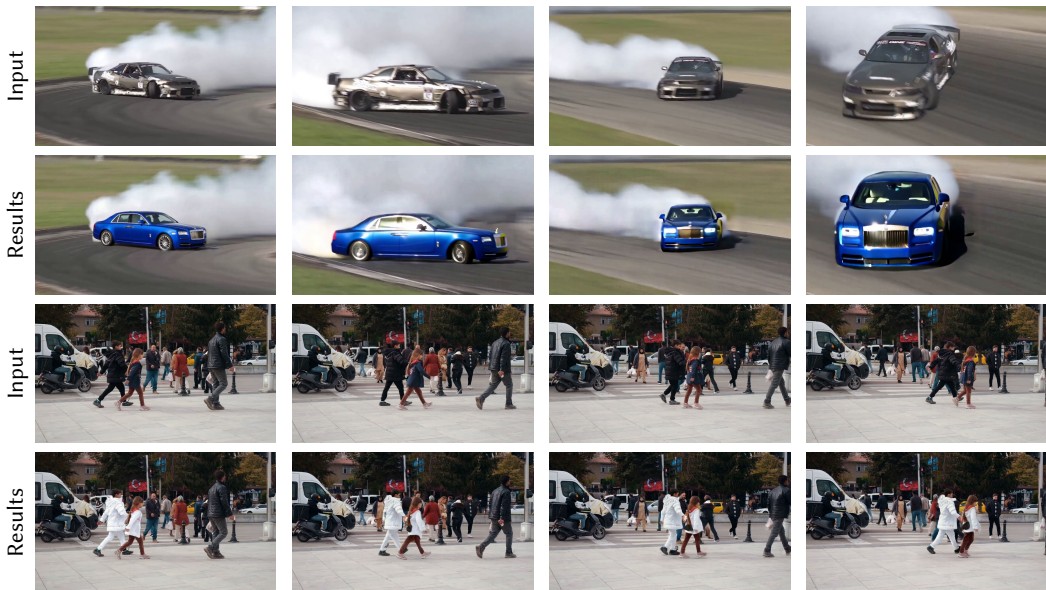

Figure 16: Diverse editing results (III).

**Instructions to Participants:** Participants were provided with the following instructions:

> "You will be shown several video sequences with different edits applied. Each sequence will contain three options, each generated using a different method. The methods will be presented in a random order to avoid bias. Your task is to choose the option that best matches the reference image in terms of visual quality and how well the edit aligns with the reference image. Pay close attention to the consistency of the edits across the frames and how naturally they interact with the environment."

After viewing each video sequence, participants were asked to answer the following two questions for each video:

1. **Reference Similarity:** Which video option best matches the reference image in terms of appearance and content?

2. **Visual Quality:** Which video option has the highest overall visual quality, considering both the foreground edit and the background consistency?

The demographic information of the participants, including their age distribution and professional background, is shown in Figure 21. This figure demonstrates the diversity of the participant pool, ensuring a wide range of perspectives in the user study.

## H    BROADER IMPACTS

The work presented in this paper introduces a flexible and controllable video editing method based on mask-aware LoRA fine-tuning. As with many advances in generative AI, our method holds both positive and negative societal implications.

Our video editing framework opens up new possibilities in creative fields, such as film production, digital art, and content creation, by enabling high-quality, real-time video edits with a high degree of control. This could streamline workflows for professionals in these industries, reducing the time and effort traditionally required for manual video editing. Additionally, our approach can be used in non-commercial domains, such as healthcare, where controllable video editing can assist in visualizing

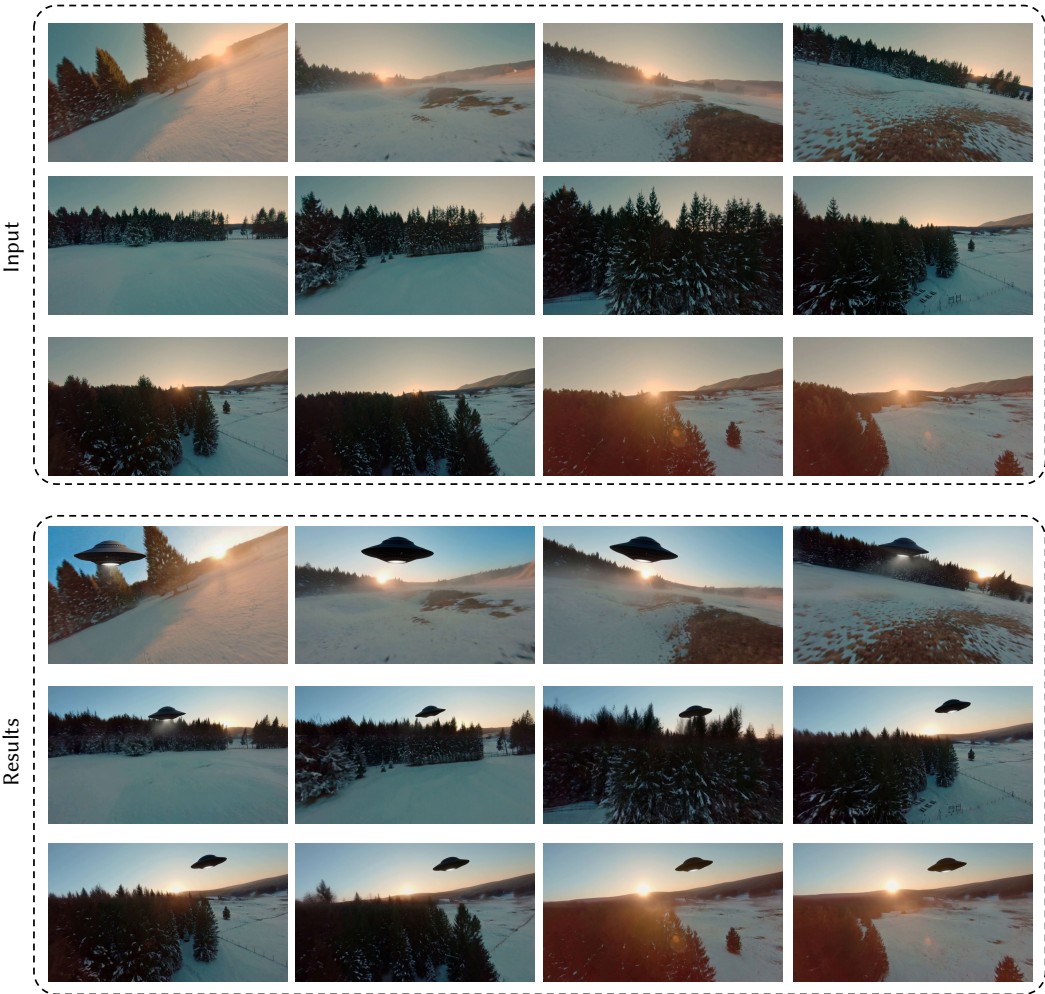

Figure 17: Diverse editing results (IV).

complex medical data, simulations, and surgical procedures. In such contexts, our method can aid in improving communication and understanding of visual information.

While the benefits are substantial, the potential for misuse of generative video models also exists. High-quality video editing tools can be misused to generate deepfakes, spreading misinformation or altering videos in malicious ways. Given the high realism of edited videos, there is a risk that such technologies could be exploited in political or social contexts, leading to challenges in verifying the authenticity of media. This raises concerns about privacy, security, and trust in video-based content.

To mitigate these risks, we advocate for responsible usage guidelines and the development of safety measures for content creation tools powered by generative models. For example, platforms using such technology could implement robust verification processes for user-generated content. Additionally, we encourage future research into model interpretability and the development of tools to detect manipulated media, ensuring that generated content is easily distinguishable from original footage.

# I  THE USE OF LARGE LANGUAGE MODELS

During the preparation of this manuscript, we utilized a large language model (LLM) to aid in polishing the writing. Specifically, it was employed to improve grammar, correct spelling, enhance

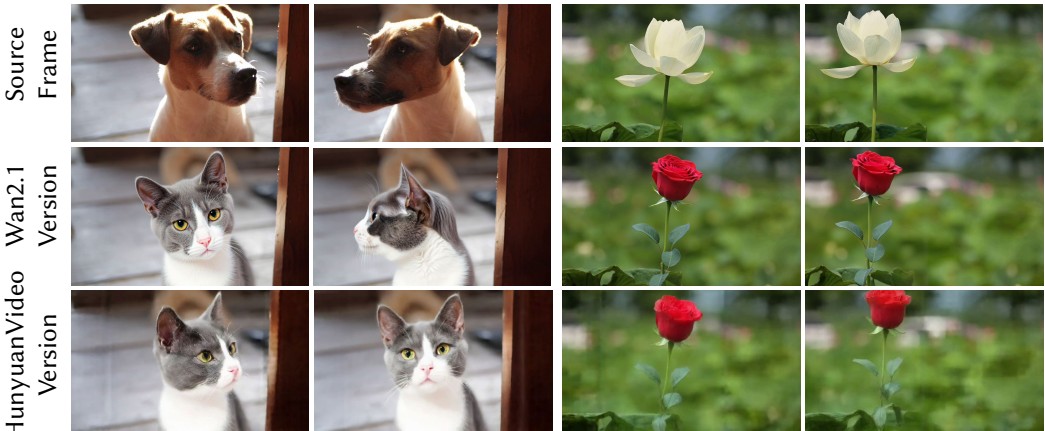

Figure 18: Results of our method applied to Wan2.1-I2V and Hunyuan Video-I2V.

clarity, and ensure the conciseness of the text. It also assisted in refining sentence structures to better adhere to a formal academic style. All scientific contributions, including the core methodology, experimental design, results, and their interpretation, were solely conceived and articulated by the human authors. The authors have thoroughly reviewed and edited the final version of the text and take full responsibility for all content presented in this paper.

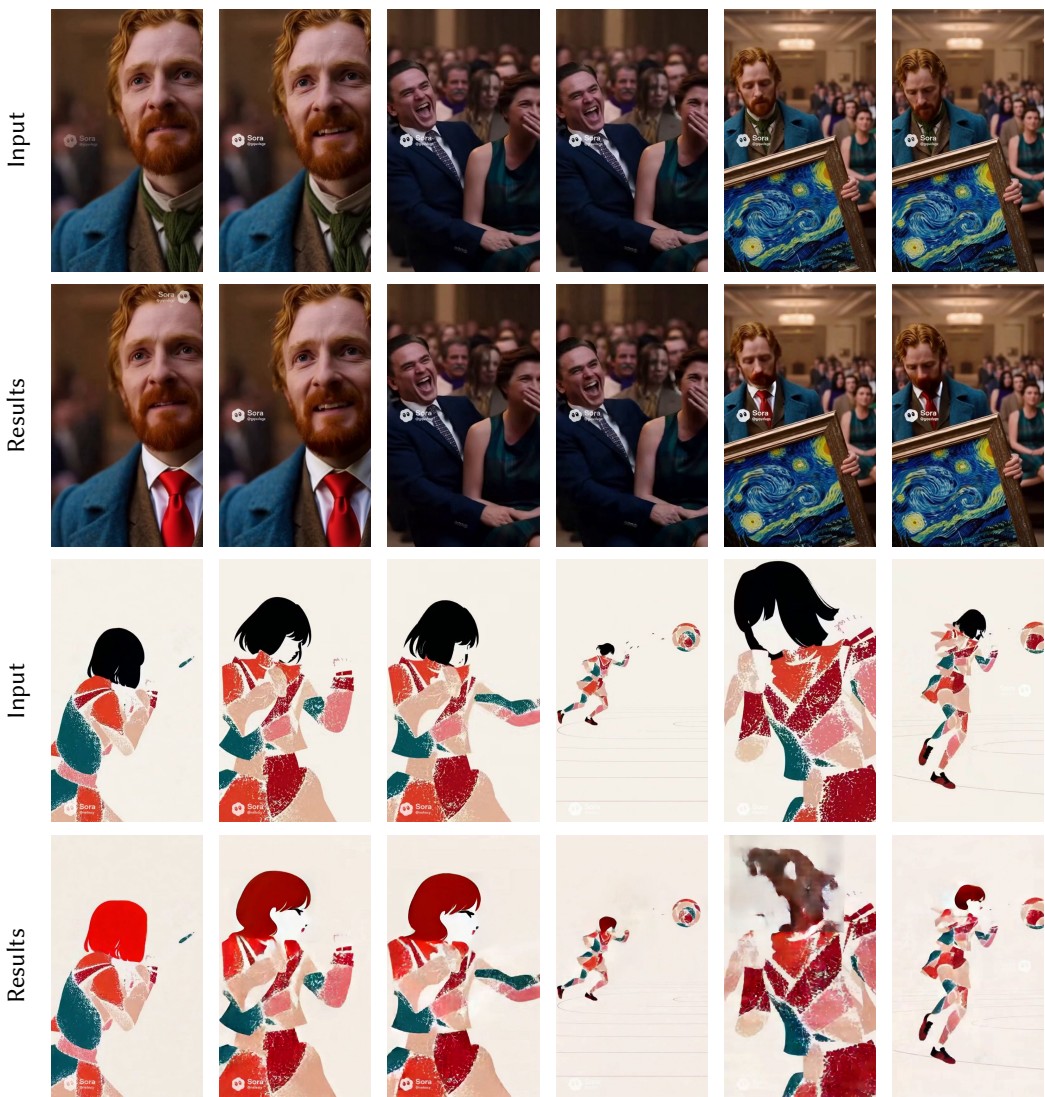

Figure 19: Failure cases. Top Rows: While the primary edit (suit and tie) is applied successfully, the model fails to preserve the fine-grained text characters in the watermark, leading to garbled glyphs. Bottom Rows: When modifying specific attributes (changing hair color to red) in scenarios with rapid and abstract motion, the method may struggle to align the new appearance with the original dynamics, resulting in body deformation and motion artifacts.

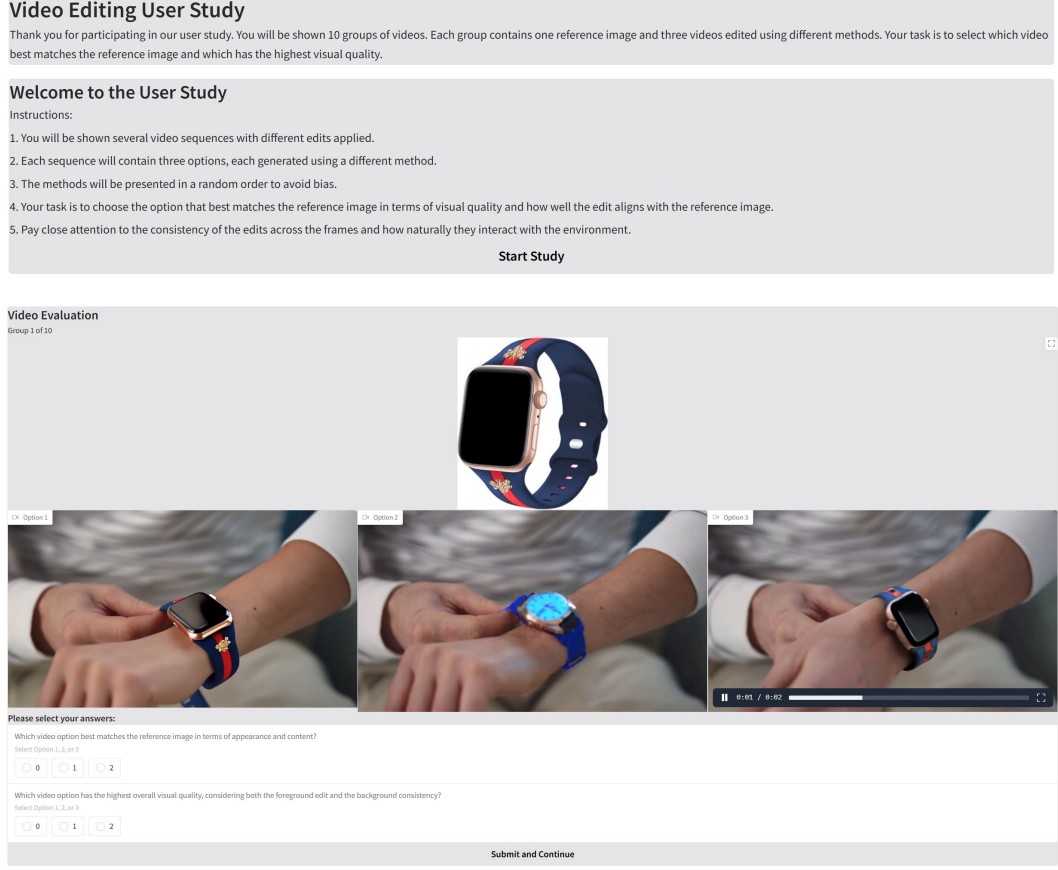

Figure 20: Screenshot of the user study interface.

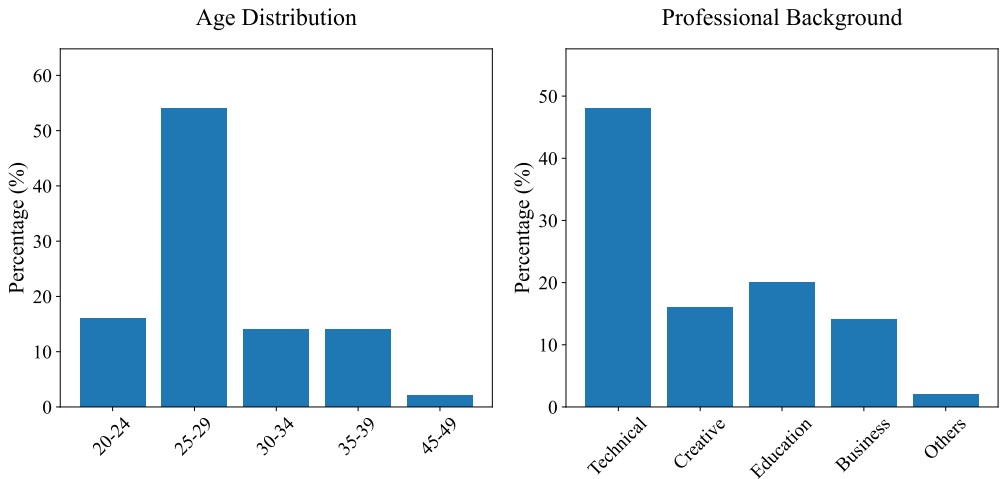

Figure 21: Demographic information of the participants in the user study.

