# OpenReview forum: "Controllable First-Frame-Guided Video Editing via Mask-Aware LoRA Fine-Tuning"
_ICLR.cc/2026/Conference — ICLR 2026 Poster_

### Official Review · Reviewer_636m · 2025-10-30

**Soundness:** 3
**Presentation:** 3
**Contribution:** 3
**Rating:** 6
**Confidence:** 5

**Summary:**

This paper proposes a video editing method that enhances first-frame-guided approaches with more granular temporal control. The core contribution is a mask-aware LoRA fine-tuning strategy for pre-trained Image-to-Video models. This technique uses a spatiotemporal mask to teach the model to selectively preserve background content while generating new content in specified regions. This dual approach allows the model to learn consistent motion from the source video and new appearances from user-provided reference frames, enabling complex edits like a flower blooming into a different color.

**Strengths:**

- Video results are pretty good (videos demonstrated in the supplementary and the demo in the supplementary showcase).

- Application-wise it's interesting and definitely be useful. However, still, efficiency is a key problem.

- Clearly outperforms previous approaches in terms of both quantitative and qualitative metrics.

**Weaknesses:**

- The primary limitation is the need of fine-tuning per-video= The method requires 100 training steps to learn motion and potentially another 100 steps to learn new appearance (Sec 4.1). While this is more efficient than full model training, it is still a significant computational step for every single video edit compared to zero-shot or inference-time methods. How do you compare it in terms of computataional efficiency? I think putting a comparison in term of run-time memory give some insights.

- The method relies on a precise "spatiotemporal mask" to separate the edited region from the background (Sec 3.3.1). The paper does not specify how this mask is acquired. Does this require the user to perform manual video segmentation for all frames? If so, this is still a requirement that would make the method impractical for most users. If the mask is generated automatically, its quality would be critical to the final edit. What could be the possible ways to automatically get these masks?

- The full-frame method requires 20GB of GPU VRAM (Sec 4.1), which is inaccessible to many users. The "low-cost training strategy" (Appendix C) is a good alternative. Can you elaborate on this, for example what ways can be done to even make it more efficient?

**Questions:**

See weaknesses, but my main questions are based on its practical applicability and efficiency.

---

> ### Author Response · Authors · 2025-11-23
> **Responses to Reviewer 636m**
>
> Thanks for the reviewer's efforts and appreciation of our framework's effectiveness. We address the main concerns below.
>
> **W1. Putting a comparison in term of run-time memory.**
>
> We thank the reviewer for the insightful suggestion to compare run-time memory, which provides a perspective on hardware accessibility. We benchmarked the run-time memory of our method against inference-based baselines. All experiments were conducted on a single NVIDIA RTX 4090 GPU, processing videos with 49 frames at a resolution of $832 \times 480$.
>
> | Method | AnyV2V | Go-with-the-flow | Ours (Standard) | **Ours (Low-Cost)** |
> | :--- | :---: | :---: | :---: | :---: |
> | **VRAM** | 13,680 MB | 4,398 MB / 21,512 MB* | 21,522 MB | **7,617 MB** |
>
> \* *Note: For Go-with-the-flow, 4,398 MB is the consumption during Diffusion Sampling, while 21,512 MB is the peak consumption during VAE Decoding.*
>
> As shown in the table, although our standard training requires ~20 GB, our **"Low-Cost Strategy"** (Appendix C) effectively reduces the training VRAM. This optimization brings our memory footprint to a level comparable with inference-only methods. This confirms that despite the need for fine-tuning, our framework remains deployable on the same class of consumer hardware as inference-only baselines.
>
> **W2. Mask Acquisition.**
>
> We appreciate the reviewer’s query regarding the practicality of mask acquisition. To clarify, our method does not require laborious frame-by-frame manual segmentation. Instead, we employ a fully automated workflow using the Segment Anything Model 2 (SAM2), initialized simply by sparse positive and negative clicks on the first frame. A user-friendly GUI for this purpose was included in the supplementary material of our initial submission, and we have now added screenshots illustrating this workflow in the **updated Appendix A** to further demonstrate its efficiency.
>
> Regarding the concern that mask quality might be a critical bottleneck, we actually observe that pixel-perfect precision is often unnecessary and can be restrictive for generative editing. Since the goal is to alter an object's appearance, using a strictly tight mask would force the new generation to adhere rigidly to the original silhouette, limiting the natural evolution of the edit. Therefore, we deliberately convert the automated segmentation into a "loose" bounding box to provide a spatial buffer for these changes. Our ablation study in the **updated Appendix B.2** confirms that this loose masking strategy yields better visual integration than precise segmentation.
>
>
> **W3. Elaborating on the Low-Cost Training Strategy**
>
> We appreciate the opportunity to elaborate on our "Low-Cost Strategy" (Appendix C). This strategy is **a training-time modification** that does not alter the inference pipeline. The original 20GB VRAM requirement primarily stems from processing the full 49-frame sequence simultaneously during training. Our approach circumvents this by splitting the training video into shorter, overlapping sliding windows. Specifically, our optimized low-cost implementation utilizes **9-frame segments**, updating the LoRA weights based on these local clips rather than the entire sequence at once.
>
> Our approach leverages the insight that **Motion LoRA primarily learns local dynamics**. Since complex motions can be decomposed into continuous short-term patterns, training on 9-frame clips is sufficient to capture the necessary kinematics. Crucially, this does not compromise global consistency because of our mask-aware conditioning. Since the unmasked background is fed as a visible input context, the model effectively learns "how the foreground moves *relative* to the fixed background" within each local window. This strong contextual anchoring ensures that the learned motion remains coherent and aligned with the environment when the full video is reassembled. We have included more visual results generated using this low-cost strategy in the **updated Appendix C.1** to demonstrate that the visual quality remains comparable to the full-frame training.
>
> Regarding engineering optimization, we utilize the "swap blocks" technique supported by the `diffusion-pipe` codebase. This mechanism dynamically offloads frozen base model layers to the CPU, keeping **only the trainable LoRA weights and the currently active Transformer blocks** in GPU memory. To demonstrate the efficiency gains, we benchmarked the peak VRAM under the 9-frame segment setting with varying configurations:
>
> | `blocks_to_swap` | 0 (No Swapping) | 16 | 32 (Default) | **38 (Max Optimization)** |
> | :--- | :---: | :---: | :---: | :---: |
> | **Peak VRAM** | 25,097 MB | 16,305 MB | 9,987 MB | **7,617 MB** |
>
> By maximizing the swap rate, we achieve a minimum footprint of 7,617 MB, making the training pipeline accessible on consumer GPUs.

---

### Official Review · Reviewer_eyka · 2025-11-01

**Soundness:** 3
**Presentation:** 3
**Contribution:** 3
**Rating:** 6
**Confidence:** 4

**Summary:**

This paper proposes a controllable first-frame-guided video editing framework that adapts a pretrained image-to-video diffusion model using mask-aware LoRA. The central idea is to repurpose the I2V model’s spatiotemporal mask not just as an inference constraint but also as a learning signal that tells LoRA what to learn. The method (i) performs per-video LoRA tuning so the model learns the input video’s motion pattern while preserving unedited regions via a mask; and (ii) optionally performs a second, brief LoRA pass where edited reference frames at later timestamps teach the model the desired appearance evolution inside masked regions (e.g., a flower gradually becoming a red rose). Experiments on first-frame-guided and reference-guided editing show better results compared with AnyV2V, Go-with-the-Flow, and I2VEdit.

**Strengths:**

- **Simple and efficient idea**
Using a spatiotemporal mask as an explicit LoRA supervision signal is a clean and reusable concept that doesn’t require architectural changes. This is likely to inspire follow-up work on using conditioning signals to steer what LoRA learns. This strategy addresses two persistent failure modes in first-frame editing: (i) unwanted background drift/leakage and (ii) insufficient control of how the edited object looks as it rotates/deforms or reveals disocclusions.

- **Low engineering overhead**
Per-video LoRA is a common practice. Adding mask-aware conditioning with edited frames is easy to deploy and works across different I2V-based models.

- Qualitative results look compelling on diverse manipulations (object add/replace/style, clothing/hair edits), with visibly better background preservation and temporal coherence than the chosen baselines.

**Weaknesses:**

- **Incremental Novelty**
Per-video LoRA for motion propagation and mask-conditioned inpainting are known practices in video editing and inpainting. The novelty is mainly in using the mask to steer LoRA’s learning objective across space-time. This is elegant but not a large algorithmic leap.

- **Mask acquisition and robustness are under-specified**
It is not clear how masks are obtained for training and inference. Is it automatically calculated from the first-frame edit difference, and then propagated with optical flow/segmentation? How to handle later edited references?
Robustness to mask errors (boundary misalignment, sparsity, temporal jitter) is not evaluated. Since the method’s key promise is precise, region-specific control, an analysis with sensitivity to mask quality is critical.

The paper tackles a high-impact practical problem (controllable propagation of first-frame edits) with a minimal, effective mechanism. The idea is easy to adopt on top of modern I2V models, and qualitative outcomes are persuasive. While the novelty is incremental, the contribution is useful and timely for creators and researchers. I tend to give a positive score.

**Questions:**

See weaknesses

---

> ### Author Response · Authors · 2025-11-23
> **Responses to Reviewer eyka**
>
> Thanks for the reviewer's efforts and appreciation of our framework's simplicity and effectiveness. We address the main concerns below.
>
> **W1. Technical Novelty**
>
> **Our major innovation lies in reshaping the information dependencies that drive the learning process**, enabling us to selectively steer the model to learn either motion dynamics or visual appearance. Previous paradigms often relied on explicit attention masking or attention map modification to manage information dependencies. In contrast, we discover that the native spatiotemporal mask inherent to modern Image-to-Video (I2V) models can be repurposed to implicitly reshape information dependencies directly. Standard I2V models enforce a fixed information flow where the first frame serves as the condition guiding the sequence. Our approach redirects these dependency paths: by strategically defining which regions serve as the immutable context and which serve as the generation target, we force the optimization to rely on specific information sources to minimize the loss. For instance, when the first frame and video background are established as the context, the model is compelled to derive the foreground's motion dynamics relative to that environment. Conversely, when a single reference image background serves as the context, the dependency shifts, forcing the model to derive how the foreground's appearance should manifest within that environment setting. This mechanism allows us to dictate knowledge extraction without requiring architectural surgery. In summary, our method governs information dependencies via the native I2V mask, achieving the goal of explicit attention modification but in a simpler and more elegant manner.
>
> **W2. Mask acquisition and robustness**
>
> We appreciate the reviewer’s query regarding the practicality of mask acquisition. We employ an automated workflow using the Segment Anything Model 2 (SAM2), initialized simply by **sparse positive and negative clicks** on the first frame. A user-friendly GUI for this purpose was included in the supplementary material of our initial submission, and we have now added screenshots illustrating this workflow in the **updated Appendix A** to further demonstrate its efficiency.
>
> Regarding the concern that mask quality might be a critical bottleneck, we actually observe that pixel-perfect precision is often unnecessary and can be restrictive for generative editing. Since the goal is to alter an object's appearance, using a strictly tight mask would force the new generation to adhere rigidly to the original silhouette, limiting the natural evolution of the edit. Therefore, we deliberately convert the automated segmentation into a "loose" bounding box to provide a spatial buffer for these changes. Our ablation study in the **updated Appendix B.2** confirms that this loose masking strategy yields better visual integration than precise segmentation.

---

### Official Review · Reviewer_4aXZ · 2025-11-03

**Soundness:** 2
**Presentation:** 3
**Contribution:** 3
**Rating:** 6
**Confidence:** 4

**Summary:**

This paper addresses first-frame-guided video editing with diffusion models, proposing a training-time, mask-aware low-rank adaptation of off-the-shelf image-to-video (I2V) models. By inserting learnable LoRA modules into attention layers and conditioning the denoising network on a user-defined spatio-temporal mask, the model is taught (i) to preserve pixels where the mask is 1 and (ii) to synthesise new content where the mask is 0, either by copying motion from the source clip or by adopting appearance from an extra reference image. At inference, only the edited first frame and the same mask are needed to propagate the edit through the whole sequence. Extensive experiments against recent baselines (I2VEdit, AnyV2V, VACE, Kling1.6) show superior or comparable DEQA, CLIP-score and user-study rankings.

**Strengths:**

- Two-stage mask scheduling (motion learning → appearance learning) cleanly disentangles dynamics from look
- Solid empirical protocol: two tasks, two backbones (Wan2.1 & HunyuanVideo), three metrics plus user study, ablations for mask usage and extra reference frames.
- Equations are minimal and directly show the change of conditioning, easing reproducibility.
- Delivers a lightweight plug-in that practitioners can apply to any I2V model; potential to become a default extension for personalised video creation tools.

**Weaknesses:**

- I2VEdit already finetunes LoRA per video and uses attention masks to reduce background leakage. The paper must quantify the marginal gain of feeding the mask into the network input instead of using it only at feature level.
- 20 self-collected clips plus an undisclosed I2VEdit test set is small. No long-form videos, fast motion, or multi-person scenes are reported.
- Perfect masks are assumed. Evaluate with noisy masks (erosion / dilation, IoU = 0.80–0.90) or segmentation-model predictions.

**Questions:**

- Provide a direct ablation that keeps everything else identical except mask conditioning is missing.
- Provide frame-count vs metric curves on Benchmark to verify scalability.
- How does performance degrade under automatic segmentation errors
- Provide some failure case

---

> ### Author Response · Authors · 2025-11-23
> **Responses to Reviewer 4aXZ (Part I)**
>
> Thanks for the reviewer's efforts and appreciation of our framework's simplicity, effectiveness and reproducibility. We address the main concerns below.
>
> **W1. Marginal gain of feeding the mask into the network input instead of using it only at feature level.**
>
> We thank the reviewer for this insightful comment. The marginal gain of input-level masking lies in bridging the gap between isolation and interaction. In I2VEdit, the LoRA is fine-tuned globally, with attention masks applied merely as an inference-time constraint to preserve the background. In contrast, our approach feeds the mask into the network input, transforming the task from simple "masked generation" to "context-aware propagation". By treating the unmasked background as a visible condition during training, we enable the model to leverage its video data priors to learn the relative evolution of the edit. Effectively, the model learns to answer: 'Given this specific background, how should the local content move and interact with it?' This ensures that the edit’s evolution (e.g., motion trajectory, lighting) is generated in intrinsic consistency with the unedited surroundings.
>
> To empirically quantify this advantage, we isolated the feature-level masking strategy for direct comparison. In this configuration, the model is fine-tuned to learn global dynamics without spatial conditioning, while background preservation is enforced strictly as an inference-time intervention. Specifically, we overwrite the generated background with the source latents at each denoising step:
> $$z_{t-1} = z_{t-1}^{pred} \odot (1 - \mathbf{M}) + z_{t-1}^{src} \odot \mathbf{M}$$
>
> As shown in the table below, our input-conditional approach yields consistent gains across all key metrics compared to feature-level baseline.
>
> | Metric | CLIP Score $\uparrow$ | DEQA Score $\uparrow$ | Input Similarity $\uparrow$ |
> | :--- | :---: | :---: | :---: |
> | **Baseline (Feature-Level)** | 0.8936 | 3.5878 | 0.7402 |
> | **Ours (Input-Level)** | **0.9172** | **3.8013** | **0.7608** |
>
> We have also included visual comparisons and analysis in the **updated Appendix B.1**.
>
> **W2. Dataset Diversity.**
>
> We have included results for long-form, fast-motion, and multi-person scenarios in the **updated Appendix D**. Our method handles these dynamics by steering the pre-trained I2V model's inherent motion priors rather than learning them from scratch. For long-form videos, we address frame limits via a sequential strategy: dividing the video into segments (e.g., 49 frames) and using the last edited frame of the previous segment to condition the next, thereby extending the processing capability to longer sequences.
>
> **W3. Perfect masks?**
>
> To clarify the details of our mask acquisition, we do not rely on "perfect" masks. We utilize the Segment Anything Model 2 (SAM2) to obtain an automated segmentation, which is then converted into bounding box. This use of a "loose" mask is a deliberate design choice: since video editing often involves altering the object's appearance, the generated entity requires a spatial buffer to undergo necessary contour variations. A strictly tight mask would excessively constrain the generation, forcing the new object to adhere to the original silhouette. In the **updated Appendix B.2**, we compared our bounding box strategy against a "noisy" mask (simulated by downsampling the segmentation to a $7 \times 7$ grid and upsampling) and the raw, tight SAM2 mask.

---

> ### Author Response · Authors · 2025-11-23
> **Responses to Reviewer 4aXZ (Part II)**
>
> **Q1. Provide a direct ablation that keeps everything else identical except mask conditioning is missing.**
>
> For the motion learning stage, this ablation is presented as the "w/o Mask" setting in Figure 6 of the main paper, demonstrating that removing the mask leads to undesirable background drift. Additionally, for the appearance learning stage, we have included a corresponding ablation in the **updated Appendix B.3**. The results show that without our proposed mask scheduling, the I2V model fails to effectively learn the target appearance from the single reference image. This confirms that the mask functions as an explicit command, instructing LoRA to switch its focus to appearance acquisition rather than motion reconstruction.
>
>
> **Q2. Provide frame-count vs metric curves on Benchmark to verify scalability.**
>
> We have included frame-count vs. metric curves in the **updated Appendix B.4**. These results demonstrate that our mask-guided LoRA is robust to variations in sequence length.
>
> **Q3. How does performance degrade under automatic segmentation errors?**
>
> Please refer to our detailed response in **W3. Perfect masks?**.
>
> **Q4. Provide some failure case.**
>
> We have included visual examples of failure cases in the **updated Appendix F**. We observe specific limitations in text preservation, particularly with fine-grained details like watermarks. Even with mask constraints, high-frequency symbols often degrade, likely due to the compression loss of the underlying VAE. Additionally, limitations arise in complex motion preservation. In scenarios featuring rapid or highly stylized abstract motion, accurately decoupling the appearance from the original dynamics becomes challenging, potentially resulting in unnatural deformations or disruptions to the structural integrity of the motion sequence.

---

### Official Review · Reviewer_bgBu · 2025-11-03

**Soundness:** 2
**Presentation:** 2
**Contribution:** 2
**Rating:** 4
**Confidence:** 4

**Summary:**

This paper proposes a method for controllable video editing by fine-tuning a pre-trained Image-to-Video (I2V) model using a mask-aware LoRA (Low-Rank Adaptation) strategy. The core idea is to use a spatiotemporal mask during LoRA tuning to guide the model in two ways: 1) to preserve or generate content in specific regions, and 2) to learn either motion from the source video or appearance from a reference image. The method is positioned as an improvement over first-frame-guided editing, offering more granular control over the temporal evolution of edits. Experiments show favorable results against several state-of-the-art baselines in both qualitative and quantitative evaluations.

**Strengths:**

The proposed framework is relatively simple, does not require architectural changes to the base model, and offers a flexible pipeline for various editing tasks (object replacement, addition, style transfer). The inclusion of a low-memory training strategy in the appendix is a practical consideration.

The paper provides a thorough experimental section, including comparisons with both first-frame-guided and reference-guided methods, quantitative metrics, ablation studies, and a user study.

**Weaknesses:**

Limited Technical Novelty: The core technical components—using I2V models, LoRA fine-tuning, and spatiotemporal masks—are not novel individually. The contribution lies primarily in their specific combination and application to this problem. However, the approach feels like a natural and incremental extension of existing "tune-a-video" and first-frame-guided paradigms. The paper does not sufficiently establish a significant technical leap over the current state-of-the-art. The method's reliance on per-video fine-tuning, while effective, is also a limitation shared with several prior works.

**Questions:**

No Questions

---

> ### Author Response · Authors · 2025-11-23
> **Responses to Reviewer bgBu**
>
> Thanks for the reviewer's efforts and appreciation of our framework's simplicity, flexibility, practical consideration and thorough experimental results. We address the main concerns below.
>
> **W1. Technical Novelty**
>
> Controllable first-frame-guided video editing requires the model to extract motion dynamics from the input video while preserving the background, and simultaneously learn target appearance from additional reference images. Previous paradigms such as existing "tune-a-video" and first-frame-guided paradigms attempted to achieve this via architectural heuristics which involved isolating spatial layers to learn appearance and temporal layers to preserve motion. **However, these approaches are becoming obsolete as state-of-the-art foundation models like Wan2.1 and HunyuanVideo evolve toward Full 3D Attention mechanisms**. In these unified architectures, spatiotemporal tokens are processed jointly, meaning that distinct parameter groups for motion or appearance no longer exist. To address this, our method capitalizes on the native masking mechanisms inherent to modern Image-to-Video architectures. By repurposing this built-in capability as a dynamic supervision signal, we proactively introduce order into the optimization, allowing us to program the LoRA to learn motion dynamics and target appearance in separate stages.
>
> Fundamentally, **our innovation lies in reshaping the information dependencies that drive the learning process**. Previous paradigms often relied on explicit attention masking or attention map modification to manage these dependencies. In contrast, we discover that the native spatiotemporal mask inherent to modern Image-to-Video (I2V) models can be repurposed to implicitly reshape these dependencies directly. Standard I2V models enforce a fixed information flow where the first frame serves as the condition guiding the sequence. Our approach redirects these dependency paths: by strategically defining which regions serve as the immutable context and which serve as the generation target, we force the optimization to rely on specific information sources to minimize the loss. For instance, when the first frame and video background are established as the context, the model is compelled to derive the foreground's motion dynamics relative to that environment. Conversely, when a single reference image background serves as the context, the dependency shifts, forcing the model to derive how the foreground's appearance should manifest within that environment setting. This mechanism allows us to dictate knowledge extraction without requiring architectural surgery.

---

### Author Response · Authors · 2025-12-01
**Summary of Author Responses**

Dear Area Chair,

We understand the challenges of the current review process and appreciate your time in evaluating our work. We have provided detailed responses and uploaded a revised PDF containing updates that address all concerns raised by the reviewers. We summarize the key resolutions below:

**1. Clarified: Technical Novelty (Addressing Reviewer bgBu & eyka)**
*   **Key Insight:** Previous methods (e.g., Tune-A-Video and I2VEdit) relied on isolating separate **temporal layers** to disentangle motion. However, this approach fails on modern **Full 3D Attention** models (e.g., Wan2.1 and HunyuanVideo) where motion and appearance parameters are merged. **Our innovation is replacing that "structural isolation" with "mask-guided optimization".** We repurpose the native mask as a supervision signal to force the model to decouple motion and appearance based on input context, effectively dictating information flow without needing the architectural surgery used in prior paradigms, thereby ensuring compatibility with powerful modern video diffusion models.

**2. Clarified & Validated: Mask Acquisition & Robustness (Addressing Reviewer 4aXZ, eyka & 636m)**
*   **Concern:** Questions regarding the practicality of mask acquisition and sensitivity to mask quality.
*   **Clarification:** Actually, our workflow is fully automated. Users only provide sparse clicks on the first frame to initialize SAM2, which automatically propagates the segmentation to subsequent frames. We then derive loose bounding boxes from these propagated masks. **No manual per-frame annotation or pixel-perfect segmentation is required**.
*   **Validation:** Our ablation study (revised Appendix B.2, Fig. 10) proves that using these **"loose masks"** actually yields **better** visual results than tight masks, as they provide a necessary spatial buffer for the object to evolve naturally.

**3. Demonstrated: High Applicability (Addressing Reviewer 636m)**
*   **Concern:** The reviewer requested elaboration on the practicality and our "Low-Cost Strategy".
*   **Response:** We validated our low-cost strategy with comprehensive benchmarking (revised Appendix C, Tables 4 & 5). Results confirm that this optimization dramatically **reduces VRAM from ~20 GB to 7.6 GB**. This makes our method deployable on consumer GPUs with overhead comparable to inference-only baselines.

Finally, regarding other requested experiments (e.g., scalability curves, diverse scenarios, and quantitative comparisons), we have included **ALL** of them in the revised Appendix, which further solidify the effectiveness and robustness of our approach.

Sincerely,

Authors

---

### Meta-Review · Area_Chair_y48U · 2025-12-25

**Summary:**

bgBu: "the paper has limited technical novelty."
The authors successfully argue what the novelty is, but the argument is not convincing. It is ultimately another simple modification in an area full of other similar simple modifications. This is the main limitation of the paper after the rebuttal.

4aXZ:
"The paper must quantify the marginal gain of feeding the mask into the network input instead of using it only at feature level."
The rebuttal addresses this point.

Additional experiments are suggested that the authors present in the newly uploaded appendix.

eyka:
"Incremental Novelty"
This problem has not been addressed by the rebuttal. It remains the major concern.

Additional results requested to evaluate masks. These experiments were added by the authors

636m:
"The primary limitation is the need of fine-tuning per-video"
Partially answered in the rebuttal. The memory consumption is ok, but the time overhead is significant compared to feedforward methods.

"Reliance on precice masks"
More details are provided.

Runtime memory overhead is too high:
This is explained better in the rebuttal

**Reviewer Concerns:**

I specified this above, together with the reviewer concerns.

**Reviewer Scores:**

bgBu --> 4 ( does not change score ). The criticism is valid, and the authors cannot address it.

4aXZ --> 8 (possibly increase to 8 from 6). The rebuttal seems to address all the concerns.

eyka --> 6 ( does not increase ). Some questions were answered, but novelty is inherently incremental.

636m --> 6 ( does not increase ). Some questions are answered, but the concept is inherently outdated compared to feedforward methods.

Based on the reviewer scores and the rebuttal, the paper should be accepted.

---

### Decision · Program_Chairs · 2026-01-26

Accept (Poster)